



# Fractional Brownian motion, the Matérn process, and stochastic modeling of turbulent dispersion

Jonathan M. Lilly[1], Adam M. Sykulski[2], Jeffrey J. Early[1], and Sofia C. Olhede[2]

[1]NorthWest Research Associates, PO Box 3027, Bellevue, WA, USA
[2]Department of Statistical Science, University College London, Gower Street, London WC1E 6BT, UK

*Correspondence to:* Jonathan Lilly (lilly@nwra.com)

**Abstract.** Stochastic process exhibiting power-law slopes in the frequency domain are frequently well modeled by fractional Brownian motion (fBm). In particular, the spectral slope at high frequencies is associated with the degree of small-scale roughness or fractal dimension. However, a broad class of real-world signals have a high-frequency slope, like fBm, but a plateau in the vicinity of zero frequency. This low-frequency plateau, it is shown, implies that the temporal integral of the

process exhibits *diffusive* behavior, dispersing from its initial location at a constant rate. Such processes are not well modeled by fBm, which has a singularity at zero frequency corresponding to an unbounded rate of dispersion. A more appropriate stochastic model is a much lesser-known random process called the Matérn process, which is shown herein to be a *damped* version of fractional Brownian motion. This article first provides a thorough introduction to fractional Brownian motion, then examines the details of the Matérn process and its relationship to fBm. An algorithm for the simulation of the Matérn process in

$O(N \log N)$ operations is given. Unlike fBm, the Matérn process is found to provide an excellent match to modeling velocities from particle trajectories in an application to two-dimensional fluid turbulence.

## 1 Introduction

Fractional Brownian motion (fBm), introduced by Mandelbrot and Van Ness (1968), is a canonical stochastic process finding wide-ranging applications in fields as diverse as oceanography (Osborne et al., 1989; Sanderson et al., 1990; Sanderson and

Booth, 1991; Summers, 2002), geophysics (Molz et al., 1997), finance (Rogers, 1997), and many others. The essential features of this process are its *self-similar* behavior—meaning that magnified and rescaled versions of the process appear statistically identical to the original—together with its *nonstationarity*, implying a never-ending growth of variance with time. Two other properties of fBm are its degree of small-scale roughness or *fractal dimension* (Mandelbrot, 1985; Falconer, 1990), and the nature of its long-term *memory* or *long-range dependence* (Beran, 1992, 1994). As pointed out by Gneiting and Schlather

(2004), the self-similarity of fractional Brownian motion links the very small and the very large temporal scales behavior together, such that its memory, fractal dimension, and self-similarity aspect ratio are all controlled by the same parameter. These, in turn, are all connected to the slope of the spectrum in the Fourier domain, where fBm exhibits a simple power-law behavior.





One important property that cannot be captured by fractional Brownian motion is the tendency for a process to *diffuse*, or disperse from an initial location at a uniform rate. In the fluid dynamics literature (e.g. Davis, 1983; LaCasce, 2008), it is known that the zero-frequency value of the spectrum of a process quantifies the dispersive tendency of the temporal integral of that process. This recognition leads to a classification of processes, proposed here, based on their spectral value at zero frequency.

We refer to random processes as *diffusive*, *subdiffusive*, or *superdiffusive*, depending on whether the spectral value is finite and nonzero, zero, or unbounded, respectively. This quality of "diffusiveness" will be shown to be related to, but distinct from, the more familiar classification of processes as short-memory or long-memory depending on the long-time behavior of their autocovariance (Beran, 1992, 1994; Gneiting and Schlather, 2004). Fractional Brownian motion is found to be superdiffusive, and is associated with a diffusivity that tends to increase without bound.

A particular application is the stochastic modeling of velocities obtained from particle trajectories in fluid flows. In the field of oceanography, one of the main windows into studying the physics of the ocean circulation consists of position data from instruments that drift freely with the currents (Rupolo et al., 1996; Rossby, 2007; Lumpkin and Pazos, 2007). Similarly, numerical models of fluid systems are frequently analyzed by examining the motion of particles carried with the flow (Pasquero et al., 2002; Veneziani et al., 2005a; Lilly et al., 2011). Such position records are known as *Lagrangian trajectories*, on account

of the moving frame of reference associated with the particles or instruments. One thread of research attempts to predict Lagrangian statistics based on dynamical assumptions (e.g. Griffa, 1996; Majda and Kramer, 1999; Berloff and McWilliams, 2002; Veneziani et al., 2005a; Majda and Gershgorin, 2013). Here, we instead try to identify the simplest stochastic model that can explain the major observed features, leaving the connection to the equations of motion to the future. Velocities from Lagrangian trajectories are found (e.g. Rupolo et al., 1996) to exhibit power-law behaviors at high frequencies, and indeed

fractional Brownian motion has been suggested as a stochastic model (Osborne et al., 1989; Sanderson et al., 1990; Sanderson and Booth, 1991; Summers, 2002). Yet a primary characteristic of these trajectories is their tendency to diffuse at a uniform rate at long times (Taylor, 1921; Davis, 1983; LaCasce, 2008; Koszalka and LaCasce, 2010), a feature that fBm cannot capture.

A type of random process having a sloped spectrum that matches fBm at high frequencies, but that takes on a constant value in the vicinity of zero frequency, exists and is known as the *Matérn* process (Matérn, 1960; Guttorp and Gneiting, 2006). The

same process has been referred to occasionally as the *fractional Ornstein-Uhlenbeck* process (Wolpert and Taqqu, 2005; Lim and Eab, 2006), because it also generalizes the well-known Ornstein-Uhlenbeck process (Uhlenbeck and Ornstein, 1930) to fractional orders. A multivariate version of the Matérn process is broadly used for spatial statistics in various fields (Goff and Jordan, 1988; Handcock and Stein, 1993; Gneiting et al., 2010; Lindgren et al., 2011; Schlather, 2012). Yet despite the appeal of its generality, the Matérn process appears in only a handful of papers in the time series literature (Wolpert and Taqqu, 2005;

Lim and Eab, 2006; Li et al., 2010; Hartikainen and Särkkä, 2010; Sykulski et al., 2016). In fluid dynamics, the only instance we have located is an application to atmospheric wind tunnel data by Von Karman (1948), as pointed out by Guttorp and Gneiting (2006).

The purpose of this paper is to investigate the theoretical properties of the Matérn process, in particular its relationship to fractional Brownian motion, and to establish the practical importance of this under-appreciated process for modeling time

series that exhibit the fundamental phenomenon of diffusion. On the theoretical side, the Matérn process is seen to be a *damped*



version of fractional Brownian motion, in the same way that the Ornstein-Uhlenbeck process is a damped version of standard Brownian motion. A simple generalization of the Matérn process that incorporates a uniform rotation rate to is shown to describe a *forced/damped fractional oscillator*. On the practical side, we find the Matérn process to be an excellent match for Lagrangian velocity spectra from a numerical simulation of two-dimensional turbulence, a classical system in fluid dynamics that has been the subject of a large number of studies, (e.g. Lin, 1972; McWilliams, 1990a; Dritschel et al., 2008; Bracco and McWilliams, 2010; Kadoch et al., 2011; Scott and Dritschel, 2013). The Matérn process allows one to simultaneously vary the values of the three most important properties of Lagrangian trajectories: the kinetic energy, the degree of small-scale roughness or fractal dimension, and the long-time diffusive behavior.

A transition of the spectrum to constant values at sufficiently low frequencies is expected to be a common feature of many physical systems. Systems are often characterized by a pressure to grow—represented by a forcing—together with some drag or resistance on that growth, represented by a damping. After a sufficiently long time, the forcing and the damping equilibrate and one reaches a bounded state. This leads to the speculation that many time series that are well described as fBm over relatively short time scales may be better matched by the Matérn process over longer time scales. More generally, the Matérn process adds a third parameter to the two parameters of fBm, thus permitting a wider range of spectral forms to be accommodated. It is therefore reasonable to think that the Matérn process could be of broad interest in many areas in which fBm has already proven itself useful.

Many of the results herein may be found somewhere in the literature; the novelty and significance of this paper arise from placing these results in context. The relevant literature is vast, and the results that form this narrative are widely distributed within disparate communities. The concept of diffusivity discussed in Section 2 is well known within physics and fluid dynamics, but is largely unheard of in the time series literature. The Matérn process investigated in Section 4 is well known in spatial statistics, but not in time series or in fluid dynamics. That the Matérn process is essentially damped fractional Brownian motion, one of our main points, has already been recognized by Lim and Eab (2006), who, however, appear to have come upon the Matérn form independently, without using this name and without referencing the existing literature. Thus, the various results brought together here currently exist in such a dispersed state that the significance of combining them is not at all apparent.

The main contributions of this work are: (i) to place the Matérn process in context by understanding its relationship to fractional Brownian motion; (ii) to establish *why* the Matérn process is important for stochastic modeling of time series, geophysical time series in particular, which is its ability to simultaneously capture the effects of long time scale diffusivity and small-scale fractal dimensionality; (iii) to demonstrate its performance with an application to a classical physical system; and (iv) to accomplish these goals in a way that is accessible to a general audience.

This paper was inspired by the need to a develop a stochastic model for a particular physical application. As such, we are cognizant of the need to make stochastic modeling tools accessible to a broad audience. We have therefore endeavored to present material in a manner that is grounded in concepts from signal analysis, as this is a common language shared by many fields. A priority is placed on being self-contained, in order to avoid referring the reader repeatedly to the literature. The use of stochastic differential equations, or other more mathematical tools, is avoided unless absolutely necessary. At the same




time, we are aware of the need to maintain rigor, and have therefore sought to carefully qualify any approximate or informal statements. New results are denoted as such.

The structure of the paper is as follows. Section 2 introduces background material regarding the concept of *diffusivity* and its relationship to the spectrum, and presents a preview of the application to turbulence as a motivation. An introduction to fractional Brownian motion is presented in Section 3. The properties of the Matérn process are then investigated in Section 4. Section 5 presents a new algorithm for fast approximate numerical generation of the Matérn process, and Section 6 returns to the application with additional details. The paper concludes with a discussion.

All numerical software associated with this paper, including a script for figure generation, is distributed as a part of a freely available Matlab toolbox, as described in Appendix A. The paper includes two supplemental animations, http://www.jmlilly.net/videos/dispersionmovie.mp4 and http://www.jmlilly.net/videos/turbulencemovie.mp4.

## 2 Background and motivation

This section introduces background material on stochastic processes, and identifies the *diffusivity* as a fundamental second-order stochastic quantity. This importance of diffusivity is illustrated by briefly discussing an application to modeling particle velocities in fluid turbulence.

### 2.1 Complex notation, continuous time

In this paper, we will work with continuous-time, complex-valued processes, a choice that deserves comment. The decision to use complex-valued processes stems from the fact that the main application, to fluid dynamics, consists of analyzing trajectories that may be regarded as positions on the complex plane. For the most part, the results all apply equally well to real-valued processes. The choice to work in continuous time reflects more than convenience. In some applications, e.g. econometrics, the processes one is attempting to model are truly discrete in nature; financial transactions, for example, occur at definite times. On the other hand, in physical applications, one frequently regards the process of interest as existing continuously in time. This process, such as a fluid flow, happens to be sampled at discrete intervals, owing to the constraints of measurements with real-world instruments. For these reasons, we will work in continuous time, and discrete sampling effects will be addressed when relevant.

### 2.2 Diffusive processes

Let $r(t) = x(t) + \mathrm{i}y(t)$ be a complex-valued random process, where $\mathrm{i} \equiv \sqrt{-1}$, subject to the initial condition $r(0) = 0$. For concreteness herein, $r(t)$ will be regarded as having units of length, with $x(t)$ and $y(t)$ giving eastward and northward coordinates, respectively. Thus $r(t)$ may be regarded as the displacement, as a function of time, of a hypothetical particle from the origin. Drawing on a key concept from physics we introduce the *total* or *isotropic diffusivity* as (see e.g. Young, 1999)

$$\kappa(t) \equiv \frac{1}{4}\frac{\mathrm{d}}{\mathrm{d}t}\mathrm{E}\left\{|r(t)|^2\right\} \tag{1}$$



which quantifies the expected rate at which the particles *disperse*, or spread out, over time from an initial location. Here $E\{\cdot\}$ is the expectation operator. Note that $\kappa(t)$ is the average of the rates of dispersion in the $x$- and $y$-directions, $\kappa_x(t) \equiv \frac{1}{2}\frac{d}{dt} E\{x^2(t)\}$ and $\kappa_y(t) \equiv \frac{1}{2}\frac{d}{dt} E\{y^2(t)\}$.

If an ensemble of particles exhibits a power-law dispersion near some time $t$ with

$$E\left\{|r(t)|^2\right\} \sim t^\beta, \qquad \kappa(t) \sim t^{\beta-1} \tag{2}$$

then the local behavior is said to be *diffusive* if $\beta = 1$, *subdiffusive* if $\beta < 1$, and *superdiffusive* if $\beta > 1$. The same process may exhibit different diffusive regimes at different times, but if (2) holds in an asymptotic sense for large $t$, then the long-time limit of $\kappa(t)$ is given by

$$\kappa \equiv \lim_{t \longrightarrow \infty} \frac{1}{4}\frac{d}{dt} E\left\{|r(t)|^2\right\} = \begin{cases} 0, & \beta < 1 \\ \text{constant}, & \beta = 1 \\ \infty, & \beta > 1 \end{cases} \tag{3}$$

where the time-independent, asymptotic quantity $\kappa$ is conventionally known simply as *the diffusivity*. In the case that $\kappa$ is a nonzero constant, one has $E\left\{|r(t)|^2\right\} = 4\kappa t$, and the *expected area* enclosed by the particle ensemble grows linearly with time. Thus $\kappa$ quantifies a tendency for random fluctuations to yield systematic outward or radial motion.

The seminal work of Taylor (1921) applied the concept of diffusivity to study the random motions of macroscopic fluid particles, a usage that is now widespread in fluid dynamics (LaCasce, 2008). Here we employ the physical concept of diffusiveness to describe the long-term dispersive behavior of random processes in general, regardless of the system being represented.

For spectral analysis, it will prove more convenient to work with the first derivative of $r(t)$, denoted $z(t) = u(t) + iv(t) = \frac{d}{dt}r(t)$. In terms of $z(t)$, $r(t)$ is given by

$$r(t) \equiv \int_0^t z(\tau)\,d\tau \tag{4}$$

where the integral is interpreted as $\int_t^0 z(\tau)\,d\tau$ for $t < 0$. This definition of $r(t)$ sets the initial condition $r(0) = 0$. Because $r(t)$ is being regarded as a particle trajectory on the complex plane, $z(t)$ represents the instantaneous particle velocity.

While the diffusivity is not a recognized quantity in time series analysis, we will show that is an essential second-order descriptor, on par with the variance. If $z(t)$ is a zero-mean second-order stationary process with autocovariance function $R_{zz}(\tau)$ and Fourier spectral density $S_{zz}(\omega)$, both defined subsequently, and having variance $\sigma^2 \equiv E\left\{|z(t)|^2\right\}$, one finds

$$\sigma^2 = R_{zz}(0) = \frac{1}{2\pi}\int_{-\infty}^{\infty} S_{zz}(\omega)\,d\omega \tag{5}$$

$$\kappa = \frac{1}{4}S_{zz}(0) = \frac{1}{4}\int_{-\infty}^{\infty} R_{zz}(\tau)\,d\tau \tag{6}$$

which shows that the variance $\sigma^2$ and diffusivity $\kappa$ may be seen as time- and frequency-domain analogues of one another. Just as the variance $\sigma^2$ is given by the integral of the velocity spectrum, or the value of the autocovariance at zero, the diffusivity



$\kappa$ is the integral of the autocovariance, or the value of the spectrum at zero. As each is the zeroth-order moment in one of the two domains, they share a common footing as the two lowest-order and potentially most important second-order statistical properties of a stationary random process.

Because the diffusivity appears as a second-order descriptor of the velocity process $z(t)$, it is useful to categorize $z(t)$ according to the associated diffusivity value. For a given $z(t)$ we may *define* $\kappa$ as in (6) through the value of the spectrum at zero frequency, or equivalently, through the integral of the autocovariance. We will refer to $z(t)$ as a *diffusive process* if it is associated in this way with a non-zero and finite value of $\kappa$. Processes associated with zero values of $\kappa$ will be said to be *subdiffusive*, while those associated with unbounded values of $\kappa$ will be referred to as *superdiffusive*. Note that the diffusivity is a property of both the velocity process $z(t)$, in the zero-frequency value of its spectrum, and the trajectory $r(t)$, in its rate of dispersion. To avoid ambiguity, we will say that $z(t)$ is a diffusive *process* whereas $r(t)$ is a diffusive *trajectory*, and so forth for sub- and superdiffusive processes.

It is clear from (6) that the notion of a diffusive process is closely related to that of process *memory*, which also has to do with the integrability of the autocovariance function $R_{zz}(\tau)$. The relationship between memory and diffusiveness will be addressed later in this section. We also point out that a *diffusive process* in our terminology is distinct from the idea of a *Markov diffusion process*, which is the solution to a particular type of first-order stochastic differential equation (e.g. Metzner, 2007). As the latter usage appears to be somewhat restricted, we expect there to be little possibility of confusion.

## 2.3 Autocovariance and spectrum

The autocovariance function of a potentially nonstationary, zero-mean, complex-valued random process $z(t)$ is defined as

$$R_{zz}(t,\tau) \equiv \mathrm{E}\{z(t+\tau)\,z^*(t)\} \tag{7}$$

where the asterisk denotes the complex conjugate; note this satisfies the symmetry $R_{zz}(t,\tau) = R_{zz}^*(t+\tau,-\tau)$. If it is the case that $z(t)$ is *second-order stationary*, its second-order statistics are by definition independent of global time $t$, and the autocovariance function is then written as $R_{zz}(\tau)$. In this case one finds $R_{zz}^*(-\tau) = R_{zz}(\tau)$, and thus the autocovariance function of a stationary complex-valued stochastic process has Hermitian symmetry. Another useful property of $R_{zz}(\tau)$ is that it is *rotationally invariant* in the $x$–$y$ plane: if one rotates the coordinate system by defining $\tilde{z}(t) \equiv e^{\mathrm{i}\Theta} z(t)$ for some constant angle $\Theta$, we have $R_{\tilde{z}\tilde{z}}(\tau) = R_{zz}(\tau)$, and the autocovariance function remains unchanged.

It is well known that the autocovariance function of a complex-valued process does not completely characterize its second-order statistics (Mooers, 1973; Picinbono and Bondon, 1997; Schreier and Scharf, 2003). Additional information is contained within a second covariance function

$$C_{zz}(t,\tau) \equiv \mathrm{E}\{z(t+\tau)\,z(t)\} \tag{8}$$





which is the covariance between $z(t)$ and its own complex conjugate.[1] This quantity is known as the *relation function* (Picinbono and Bondon, 1997) or *complementary autocovariance function* (Schreier and Scharf, 2003) in the time series literature, and as the *outer autocovariance* in oceanography and atmospheric science (Mooers, 1973). Unlike the autocovariance function, the relation function changes with a coordinate rotation. With $\tilde{z}(t) \equiv e^{i\Theta} z(t)$, one finds $C_{\tilde{z}\tilde{z}}(\tau) = e^{i2\Theta} C_{zz}(\tau)$. This shows that information regarding the directionality of variability must reside in $C_{zz}(t,\tau)$ and not in $R_{zz}(t,\tau)$. If the process is isotropic, then clearly $C_{zz}(t,\tau)$ must vanish. In the present paper we are concerned with isotropic processes, and we will therefore limit our attention to $R_{zz}(t,\tau)$.

The statistical information contained in $R_{zz}(\tau)$ can be equivalently expressed in terms of its Fourier transform, the spectrum $S_{zz}(\omega)$, through the inverse Fourier relationship

$$R_{zz}(\tau) = \frac{1}{2\pi} \int\limits_{-\infty}^{\infty} e^{i\omega\tau} S_{zz}(\omega)\, d\omega. \tag{9}$$

Rather than needing to deal separately with an eastward or $u$-velocity spectrum and a northward or $v$-velocity spectrum, the spectrum of the complex-valued velocity $z(t) = u(t) + iv(t)$ compactly includes contributions due to positively-rotating circular motions $e^{i|\omega|\tau}$ for $\omega > 0$, and those due to negatively-rotating circular motions $e^{-i|\omega|\tau}$ for $\omega < 0$. For this reason $S_{zz}(\omega)$ is referred to as the *rotary spectrum* in the oceanographic and atmospheric science literature (Fofonoff, 1969; Gonella, 1972; Mooers, 1973; Emery and Thomson, 2014). Because physical processes are generally better separated in the frequency domain than in the time domain, and because the spectrum is a more straightforward quantity to estimate than is the autocovariance, we will work with the spectrum rather than the autocovariance for stochastic modeling.

It was stated earlier in (6) that for a stationary process, the diffusivity $\kappa$ is the value of the spectrum at zero frequency. This is shown as follows, by beginning with the nonstationary case. The *time-dependent* diffusivity $\kappa(t)$ can be expressed in terms of the nonstationary covariance function $R_{zz}(t,\tau)$ as

$$\kappa(t) = \frac{1}{4} \frac{d}{dt} \int\limits_{0}^{t} \int\limits_{0}^{t} E\{z(t_1)\, z^*(t_2)\}\, dt_1\, dt_2 \tag{10}$$

$$= \frac{1}{4} \frac{d}{dt} \int\limits_{0}^{t} \left[ \int\limits_{0}^{t} R_{zz}(t_2, t_1 - t_2)\, dt_1 \right] dt_2 \tag{11}$$

after substituting (4) into (1) together with (7). Applying the Leibniz rule for differentiation of an integral, in the form

$$\frac{d}{dt} \int\limits_{0}^{t} f(\tau, t)\, d\tau = f(t,t) + \int\limits_{0}^{t} \frac{\partial}{\partial t} f(\tau, t)\, d\tau \tag{12}$$

---

[1]It is considered standard that the covariance between two zero-mean complex-valued time series $a(t)$ and $b(t)$ involves a conjugation of one of the two time series, e.g. $R_{ab}(\tau) \equiv E\{a(t+\tau) b^*(t)\}$. This accounts for the conjugation in (7) and the absence of conjugation in (8). Thus, the quantity $C_{zz}(t,\tau)$ may be equivalently, but rather confusingly, denoted as $R_{zz^*}(t,\tau)$.





the expression for the time-dependent diffusivity simplifies to

$$\kappa(t) = \frac{1}{4} \int_0^t R_{zz}(t, t_1 - t) \, dt_1 + \frac{1}{4} \int_0^t R_{zz}(t_2, t - t_2) \, dt_2$$

$$= \frac{1}{2} \int_0^t \Re\{R_{zz}(t, \tau - t)\} \, d\tau \tag{13}$$

where in applying (12), $f(\tau, t)$ is taken to be the entire quantity in square brackets in (11). The second line in (13) follows from

the symmetry $R_{zz}(t, \tau) = R_{zz}^*(t + \tau, -\tau)$, with $\Re\{\cdot\}$ denoting the real part.

The time-dependent diffusivity can be understood in several different ways, see also LaCasce (2008). Substituting the definition of the autocovariance (7), the last expression in (13) becomes

$$\kappa(t) = \frac{1}{2} \int_0^t \Re\{\mathrm{E}\left[z(\tau) z^*(t)\right]\} \, d\tau \tag{14}$$

which states that the time-dependent diffusivity is the integral of the covariance between the velocity at time $t$ and the velocity

at all other times between $0$ and $t$. However, $z^*(t)$ can be pulled outside the integral, leading to

$$\kappa(t) = \frac{1}{2} \Re\left\{\mathrm{E}\left[z^*(t) \int_0^t z(\tau) \, d\tau\right]\right\} = \frac{1}{2} \Re\{\mathrm{E}\left[z^*(t) r(t)\right]\} \tag{15}$$

so that $\kappa(t)$ can equivalently be seen as the *inner product* of the velocity at time $t$ and the displacement at time $t$.

In the case that $z(t)$ is stationary, $R_{zz}(t, \tau) = R_{zz}(\tau)$, and the long-time limiting diffusivity value $\kappa$ is given by

$$\kappa = \lim_{t \longrightarrow \infty} \frac{1}{2} \int_0^t \Re\{R_{zz}(\tau - t)\} \, d\tau \tag{16}$$

$$= \lim_{t \longrightarrow \infty} \frac{1}{2} \int_{-t}^0 \Re\{R_{zz}(\tau)\} \, d\tau = \frac{1}{4} \int_{-\infty}^\infty R_{zz}(\tau) \, d\tau \tag{17}$$

after a change of variables. One may invert the inverse Fourier transform (9) to give $S_{zz}(\omega) = \int_{-\infty}^\infty e^{-\mathrm{i}\omega\tau} R_{zz}(\tau) \, d\tau$, and then $\kappa = S_{zz}(0)/4$, as claimed in (6). Thus, while diffusivity is generally thought of as a time-domain quantity, it may also be expressed in the frequency domain.

The result that the diffusivity is the zero-frequency value of velocity spectrum is not entirely new. It is implicit in a result

of Kampé de Fériet (1939), see p. 527–528 of Monin and Yaglom (2007). It is also pointed out in Davis (1983, p. 175) and is mentioned in LaCasce (2008). However, this result does not appear widely appreciated in the ocean/atmosphere literature. Within the time series literature, there does not appear to be a recognition of the potential importance of the zero-frequency value of the spectrum on account of its connection to dispersive behavior.





**Table 1.** Examples of spectra for short-and long-memory processes of subdiffusive, diffusive, and superdiffusive types. The term in the box is the Matérn process. The short-memory subdiffusive process has $\alpha > 3/2$, while the others have $\alpha > 1/2$.

| | $\kappa$ | Short Memory | Long Memory |
|---|---|---|---|
| Superdiffusive | $\infty$ | (not possible) | $\dfrac{1}{\omega^2 \left(\omega^2 + \lambda^2\right)^\alpha}$ |
| Diffusive | 1 | $\boxed{\dfrac{4\lambda^{2\alpha}}{\left(\omega^2 + \lambda^2\right)^\alpha}}$ | $\dfrac{4\Omega^2 \left(\Omega^2 + \lambda^2\right)^\alpha}{\lvert\omega - \Omega\rvert^2 \left(\lvert\omega - \Omega\rvert^2 + \lambda^2\right)^\alpha}$ |
| Subdiffusive | 0 | $\dfrac{\omega^2}{\left(\omega^2 + \lambda^2\right)^\alpha}$ | $\dfrac{\omega^2}{\lvert\omega - \Omega\rvert^2 \left(\lvert\omega - \Omega\rvert^2 + \lambda^2\right)^\alpha}$ |

## 2.4 Diffusiveness and memory

The classification of a stochastic process as diffusive, subdiffusive, or superdiffusive is related to a more familiar property, the process *memory*. If the autocovariance of a finite-variance stationary process exhibits the long-term decay

$$R_{zz}(\tau) \sim |\tau|^{-\mu}, \qquad 0 < \mu \leq 1 \qquad |\tau| \to \infty \tag{18}$$

then the process is said to be a *long-memory process* or to have *long-range dependence* (Beran, 1992, 1994; Gneiting and Schlather, 2004). A *short-memory* process is one for which the autocovariance falls off more rapidly than $|1/\tau|$, such that (18) is not satisfied. Note that the statement $R_{zz}(\tau) \sim |\tau|^{-\mu}$ means that the *magnitude* of the autocovariance decays as $|\tau|^{-\mu}$. Because the autocovariance is bounded by the variance, $R_{zz}(\tau) < \sigma^2$, condition (18) not being satisfied is equivalent to the autocovariance being absolutely integrable. Thus, short-memory processes are those for which the autocovariance function is

absolutely integrable, and long-memory processes are those for which it is not.

     The process memory is therefore a classification based on the *absolute integrability* of the autocovariance, whereas the diffusiveness is based on its *integrability*, see (6). From this one may establish that both short- and long-memory processes can be diffusive or subdiffusive, but only long-memory processes can be superdiffusive. A long-memory process has an autocovariance that is not *absolutely integrable*, whereas a diffusive process has an autocovariance that is *integrable* and that integrates

to a nonzero value. A function can be integrable but not absolutely integrable, thus a diffusive process can be long-memory. Similarly, both short-memory and long-memory processes could have autocovariances that integrate to zero, giving a subdiffusive process. However, if a function is integrable then it is also absolutely integrable, thus a short-memory process cannot be superdiffusive.

     Examples of spectra of stationary processes corresponding to different combinations of memory and diffusiveness are given

in Table 1. All of these examples are constructed by various operations on the Matérn process, which itself is an example of a short-memory diffusive process. Taking a derivative of the Matérn process multiplies the spectrum by $\omega^2$ and thus leads to





a short-memory subdiffusive process, which has finite variance provided we choose $\alpha > 3/2$. Integrating the Matérn process divides the spectrum by $\omega^2$, giving a process that is both long-memory and superdiffusive. Modulating this process by $e^{i\Omega t}$ for a nonzero frequency $\Omega$ modulates the autocovariance by $e^{i\Omega \tau}$, thus shifting the spectrum by $\omega \mapsto \omega - \Omega$. The resulting spectrum has a finite value at frequency zero but a singularity off zero, and is therefore diffusive but long-memory; we note

that this continuous-time process is related to the discrete-time Gegenbauer process (Gray et al., 1989; Baillie, 1996). Finally, differentiating the previous process causes the spectrum at zero frequency to vanish, but does not remove the singularity, leading to a long-memory subdiffusive process.

In this table, the two spectra corresponding to diffusive processes have been normalized such that $\kappa = S_{zz}(0)/4 = 1$. As described above, the classification of a process as 'diffusive' means that its spectrum takes on a finite nonzero value at zero

frequency, such that the integrated version of the process exhibits diffusive dispersion as described by (3).

## 2.5   Application to 2D turbulence

In this paper, we will be concerned with an application to the stochastic modeling of particle trajectories, and the associated velocity time series, from a numerical simulation of fluid turbulence. The system we will use, known as *forced-dissipative two-dimensional turbulence*, see e.g. Vallis (2006), generates temporally and spatially varying flows that exist purely in the

horizontal plane. This system is considered an idealized representation of turbulence in planetary fluid dynamics. Details of the numerical model, including the model equations and parameter choices, are described in Section 6.1. The simulation is carried out in a doubly periodic domain having physical dimension of $2500 \times 2500$ km, and is integrated for three years. The time series to be analyzed here are taken from 1024 particle trajectories that are tracked throughout this experiment, and that are initially uniformly distributed throughout the model grid at regular intervals.

A snapshot of the velocity field at the initial time, together with the particle trajectories from the entire simulation, is shown in Fig. 1. The quantity plotted in the left-hand panel is the current speed $|U + iV| = \sqrt{U^2 + V^2}$ at time $t = 0$, where $U = U(x, y, t)$ and $V = V(x, y, t)$ are the velocities at each point in the domain. The roughly circular regions of high-speed currents correspond to long-lived swirling structures termed *vortices* or *eddies*; their emergence is one of the defining features of two-dimensional turbulence (e.g. McWilliams, 1990a).

Trajectories from simulations such as this one can be usefully categorized into two classes: those which exhibit high-frequency oscillations characteristic of trapping within a vortex, and those which do not. Because of the long-lived nature of vortices, particles tend to persist either inside or outside of vortices for long periods of time, see Pasquero et al. (2002). These oscillatory features, which appear as broadband peaks in the spectral domain, may be identified and extracted from the time series using a different method developed by two of the authors (Lilly and Gascard, 2006; Lilly and Olhede, 2009; Lilly

et al., 2011; Lilly and Olhede, 2012); they are not the subject of the present paper. For this reason, one-half of the trajectories are discarded in order to exclude those directly effected by vortices, using a criterion described in Section 6.1, leaving 512 trajectories that will be analyzed herein. The supplementary animation turbulencemovie.mp4 presents the evolution of these 512 trajectories superimposed on the speed as in Fig. 1a.



**Current Speed**                   **Particle Trajectories**

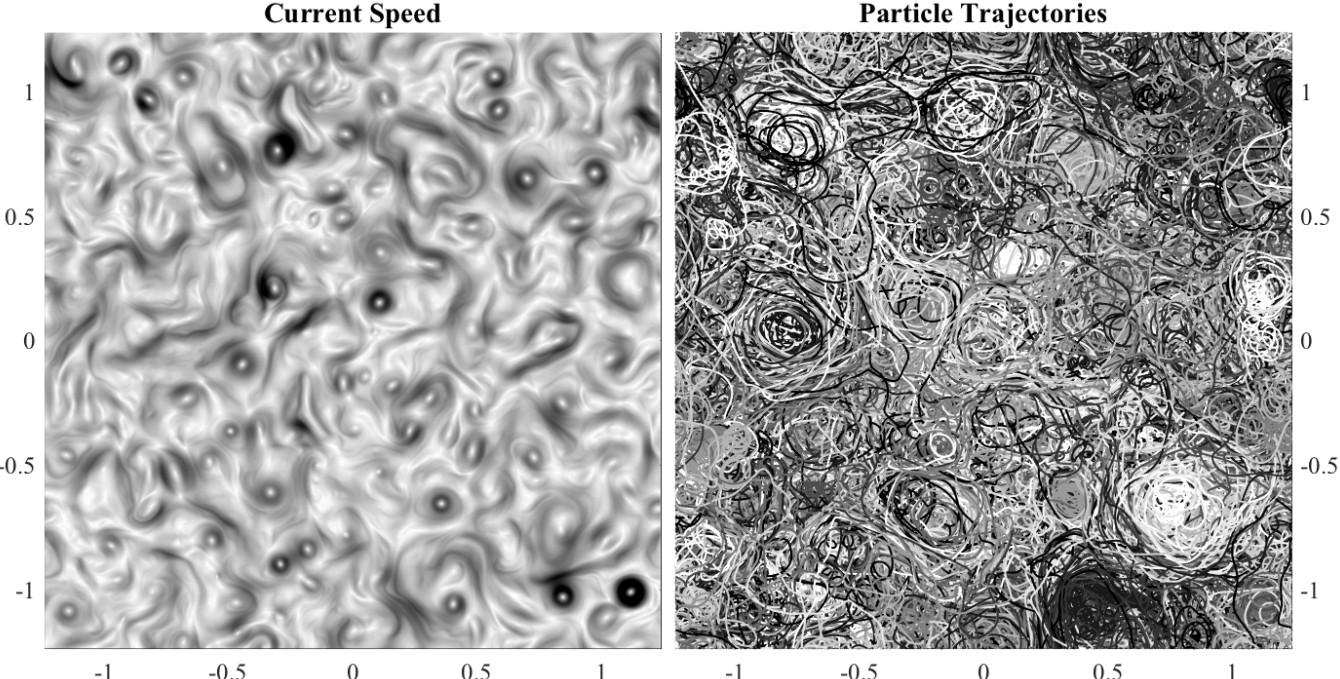

**Figure 1.** A snapshot of current speed from the turbulence simulation (left) together with 1024 particle trajectories (right). In the left panel, shading is the speed $\sqrt{U^2(x,y,t)+V^2(x,y,t)}$ at each point, with white corresponding to zero velocity and black to 18 cm s$^{-1}$. In the right-hand panel, different trajectories are represented by different shadings of gray. The physical domain size is $2500 \times 2500$ km, with the $x$- and $y$-axes in this figure given in units of 1000 km. See turbulencemovie.mp4 for an animation of this figure, in which only the 512 particles to be analyzed are shown.

These 512 "eddy-free" trajectories are also displayed Fig. 2a. Here, the position coordinates in the periodic domain have been unwrapped, and the resulting trajectories $r(t)$ offset to begin at the origin at time $t = 0$. Dispersion is then visualized by the circles, which have been drawn with radii

$$\tilde{r}_n \equiv \sqrt{\mathrm{E}\left\{|r(n\Delta t)|^2\right\}} \qquad (19)$$

at uniformly spaced time intervals $n\Delta t$, with $\Delta t$ equal to six months and $n = 1, 2, \ldots 6$. In this expression, the expectation operator is interpreted as the average over all 512 trajectories. For constant diffusivity, one expects that $\tilde{r}_n^2 = 4\kappa n\Delta t$ from (1), such that the total enclosed area increases linearly, and the radius increases as the square root of time. That the trajectories shown here are exhibiting diffusive behavior is thus indicated by the appearance of the circles in Fig. 2a, which become more closely spaced together as time increases.

The average estimated spectrum of the velocity signals $z(t) = \frac{\mathrm{d}}{\mathrm{d}t}r(t)$ corresponding to these trajectories is shown as the heavy curve in Fig. 3. Estimates of the velocity spectra have been formed for each trajectory by tapering with a lowest-order Discrete Prolate Spheroidal Sequence or "Slepian" taper (Slepian, 1978; Thomson, 1982; Park et al., 1987; Percival and

**Dispersion for: (a) Geostrophic Turbulence, (b) Matern, (c) White Noise, (d) Power-Law Process**

**Figure 2.** [See caption on next page]



**Figure 2.** [See figure on previous page] Dispersion curves for the three-year turbulence trajectories and the three different stochastic models discussed in Section 2.6. Panel (a) shows 512 "eddy-free" trajectories, chosen from a larger set of 1024 as described later in Section 6.1. All curves have been offset such that the initial points are located at the origin. Panel (b) shows realizations of a Matérn random process using parameters fit to the velocity spectra of each trajectory, and then cumulatively summed to produce a displacement, also with the initial condition at the origin. Similarly, the lower two panels show trajectories corresponding to white noise velocities (c) and velocities for a power-law process (d), the latter approximated using a Matérn process with very low damping. The stochastic velocities in (c) are chosen to match the low-frequency spectral levels of the turbulence trajectories, while in (d) they are chosen to match the high-frequency spectral slope. All trajectories in the doubly-periodic domain have been unwrapped for presentational clarity, with the gray square in each panel showing the domain size. Note that the x- and y-axes in panel (d) are a factor of one million times larger than those of the other panels, which is why the gray box is not visible. In each panel, black circles show the root-mean-square distance from the origin $\tilde{r}_n$ defined as in (19). Circles are drawn every six months, beginning at six months and ending at three years. The circles in (d) do not become closer together with increasing radius, indicating superdiffusive behavior. See dispersionmovie.mp4 for an animation of the first two panels of this figure.

Walden, 1993) having a time-bandwidth product set to a value of 10, see Park et al. (1987) or Percival and Walden (1993) for a definition of this parameter. The spectra for all 512 velocity signals are averaged together, and because there is no expected difference between clockwise and anti-clockwise velocities, the spectra for positive and negative frequencies are averaged together as well, leading to a single curve defined only for non-negative frequencies.

The velocity spectrum is observed to have three main features: an overall energy level, a high-frequency slope, and a low-frequency plateau. As shown in the preceding section, the low-frequency plateau of the velocity signals is a reflection of the diffusive behavior of the trajectories. The goal of this paper is to identify a stochastic model capable of reproducing these features, and to thoroughly understand its properties.

## 2.6    Overview of stochastic models

Consider one-, two-, and three-parameter frequency spectra having the forms

$$S_{zz}(\omega) = A^2, \quad S_{zz}(\omega) = \frac{A^2}{|\omega|^{2\alpha}}, \quad S_{zz}(\omega) = \frac{A^2}{(\omega^2 + \lambda^2)^\alpha}$$

which are taken as models for the complex velocity time series $z(t)$ from the turbulence simulation. The first type of spectrum corresponds to white noise,[2] while the second is a power-law spectrum that arises for fractional Brownian motion (Mandelbrot and Van Ness, 1968) for a certain range of $\alpha$ values; $\alpha$ is here referred to as the *slope parameter*. The third spectrum is that of

a type of random process known as a Matérn process (Matérn, 1960; Guttorp and Gneiting, 2006), which we will show to be a *damped* version of fractional Brownian motion, with $\lambda$ playing the role of an inverse damping timescale.

The form of the Matérn spectrum is fit to the velocity spectra of the turbulence trajectories, in a way that will be described in Section 6, to generate best-fit values of the three Matérn parameters for each of the 512 trajectories. The low-frequency values

---

[2]For the sake of brevity, we are discussing spectra in continuous frequency while presenting a white spectrum $S_{zz}(\omega) = A^2$ that is defined over all frequencies. This glosses over the fact that white noise is a discrete process with a spectrum that is defined only up to the Nyquist.





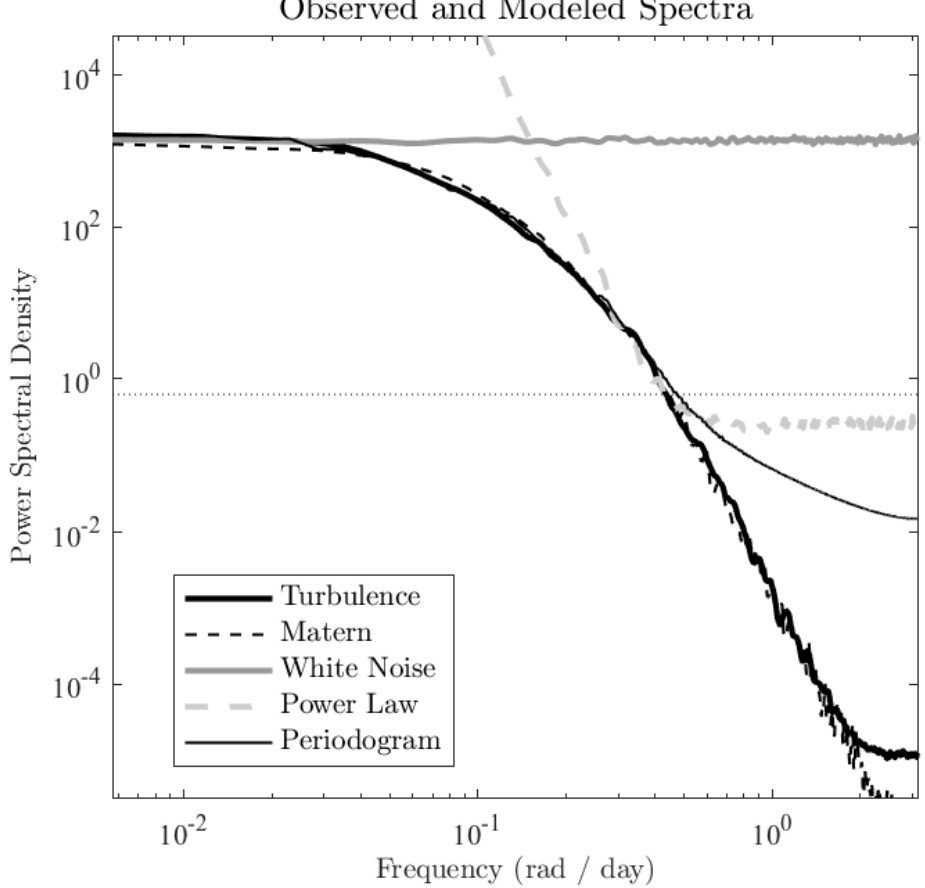

**Figure 3.** Spectra for the trajectories shown in Fig. 2. Estimated rotary spectra $S_{zz}(\omega)$ are shown for positive frequencies only, since the negative frequency side is statistically identical. The first four curves show the mean value of the estimated spectrum of each of the four sets of trajectories shown in Fig. 2, formed using a data taper as described in the text. The fifth curve, indicated by a solid line, is the mean periodogram of the turbulence trajectories, and illustrates the consequences of not accounting for spectral leakage. The dotted horizontal line marks the approximately limit of double numerical precision for the power paw process, 15 orders of magnitude below its maximum; this precision limit accounts for the flattening of gray dashed curve.

from these fits are then used to match the white noise spectrum, while the parameters for the high-frequency slopes are used to match the power-law spectrum. For each set of parameters, realizations of these three types of random processes are constructed using the best-fit parameters as described in Section 5. The spectra of the simulated trajectories are then estimated in the same manner as for the original trajectories, and shown in Fig. 3. As expected due to their construction, the white noise and power-

5 law process match only the low-frequency plateau or high-frequency spectral slope, respectively, of the original spectra. The Matérn process is seen to provide an excellent match to the observed spectra over roughly eight decades of structure.





These three different sets of random processes for the velocity time series are then cumulatively summed to form trajectories, and are compared with the original trajectories in Fig. 2; note the axes limits in Fig. 2d are a factor of one million times larger than in the other panels. The turbulence trajectories and the synthetic trajectories generated from the Matérn model are observed to be virtually indistinguishable in character. See the supplementary file dispersionmovie.mp4 for an animation of the upper
two panels of Fig. 2, showing the good agreement between the Matérn trajectories and the turbulence trajectories.

By contrast, the one-parameter and two-parameter spectral models provide poor fits to the observed trajectories, see Fig. 2c,d. The trajectories associated with the white noise velocities match the dispersion curves closely, but the trajectories are far too rough in appearance. When set to match the high-frequency spectral slope and thus the trajectory behavior at small scales, the power-law model for velocity spectra yields trajectories with a vastly incorrect range, too high a degree of smoothness at the
large scale, and dispersion characteristic of a continually increasing diffusivity.

Thus, the white noise model is able to correctly match the large scale, low-frequency component of the velocity spectra that accounts for the diffusive behavior of the trajectories. The power-law model is able to correctly match the high-frequency component of the spectrum that sets the small-scale roughness. The Matérn spectrum allows one to match both. This provides a compelling example that motivates examining the Matérn process in more detail.

## 3   Fractional Brownian motion

This section reviews the properties of fractional Brownian motion, focusing on the central importance of the spectrum. With a few noted exceptions, this section presents material that is already known in the literature.

### 3.1   Spectrum

As described in the Introduction, many real-world processes are found to exhibit power-law behavior over a broad range of
frequencies. For a range of spectral slopes, the power-law spectrum corresponds to that of a random process called *fractional Brownian motion* (fBm), introduced by Mandelbrot and Van Ness (1968). The spectrum of fBm is given by (Flandrin, 1989; Solo, 1992)

$$S_{zz}^{fBm}(\omega) = \frac{A^2}{|\omega|^{2\alpha}}, \qquad 1/2 < \alpha < 3/2 \tag{20}$$

where $\alpha$ will be called the *slope parameter*, and with $A$ setting the spectral level. Fractional Brownian motion is a gener-
alization of classical Brownian motion—corresponding to the case $\alpha = 1$ and therefore to an $\omega^{-2}$ spectrum—for which the slope parameter can take a range of non-integral values. It is clear that a process having a spectrum proportional to $|\omega|^{-2\alpha}$ for $\alpha > 1/2$ will be singular at zero, and will integrate to an infinite value, thus possessing neither a finite diffusivity nor a finite variance. Both the variance and the diffusivity of fBm will be found to increase without bound.

Examples of complex-valued fractional Brownian motion are shown in Fig. 4. Here nine curves are shown for nine different
values of $\alpha$, varying from just greater than $1/2$ to just less than $3/2$. The increase in the degree of roughness as $\alpha$ decreases, and the spectral slope becomes less steep, is readily apparent in the figure. Because we are considering that $z(t)$ represents a





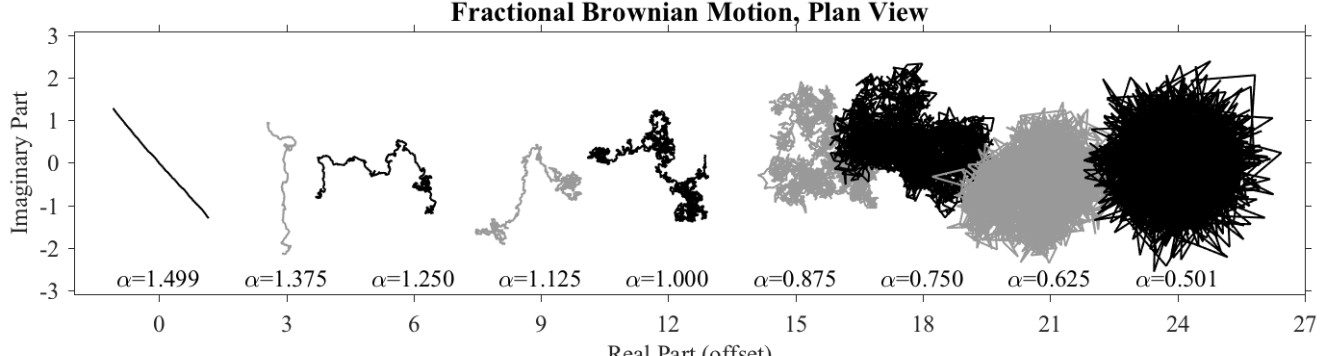

**Figure 4.** Plan view of realizations of complex-valued fractional Brownian motion $z(t)$, for nine different values of the slope parameter $\alpha$, as indicated in the legend. All nine processes have been set to have unit variance, and the real part of each curve is offset by a value of $-3$ from that for the next lower value of $\alpha$. The smallest value of $\alpha$, corresponding to the least smooth process, is at the right. The data aspect ratio is equal between the real and imaginary parts.

velocity $z(t) = u(t) + iv(t)$, this figure shows plots of $u(t)$ versus $v(t)$, as opposed to the trajectories that would arise from temporally integrating these quantities.

The main goal of this section is to utilize fBm to understand the implications of the slope parameter $\alpha$. It will be found that for fractional Brownian motion, $\alpha$ has several intuitively distinct but partly corresponding interpretations: it is directly linked to the temporal decay of the autocovariance function; it controls the aspect ratio of rescaling for self-similar behavior; it sets the *fractal dimension* or *degree of roughness*; and it determines the degree of *persistence* or *anti-persistence* of a differenced version of the process, see Appendix C. Note that in the fBm literature, the slope parameter $\alpha$ is conventionally replaced with $H = \alpha - 1/2$, referred to as the *Hurst parameter*, with $0 < H < 1$, in terms of which the fBm spectrum is given by $S_{zz}^{fBm}(\omega) = A^2/|\omega|^{2H+1}$.

There are compelling reasons to work with the slope parameter $\alpha$ rather than the Hurst parameter $H$. While the one could describe the spectral slope in the vicinity of any frequency, the Hurst parameter is, strictly speaking, a measure of the long-time process range or memory. That is, $H$ is a *limiting* quantity pertaining to the behavior of the process at very large time scales. As pointed out by Gneiting and Schlather (2004), the self-similarity of fBm implies that the large-scale behavior (memory) and small-scale behavior (fractal dimension) must be linked. However, for stochastic processes more generally, no such link is required. The spectral slope is therefore more appropriate when showing the connection of fBm to its damped version, the Matérn process, which is a short-memory process. Furthermore, because the appearance of the Matérn process as damped fractional Brownian motion is most clear in the frequency domain, it is sensible to work with a parameter that makes the spectral form simple.





## 3.2 The fBm autocovariance function

Fractional Brownian motion is defined in terms of a stochastic integral equation, which will be presented later in this section. This stochastic integral equation leads to an autocovariance function given by (Mandelbrot and Van Ness, 1968)

$$R_{zz}^{fBm}(t,\tau) = \mathrm{E}\{z(t+\tau)\,z^*(t)\} = \frac{V_\alpha}{2}\,A^2\left[|t+\tau|^{2\alpha-1} + |t|^{2\alpha-1} - |\tau|^{2\alpha-1}\right] \tag{21}$$

where $V_\alpha$ is a normalizing constant defined shortly. The exponent $2\alpha-1$ varies from a minimum of just greater than zero to a maximum of just less than two as $\alpha$ varies from just greater than 1/2 to just less than 3/2. Thus the dependence of $R_{zz}^{fBm}(t,\tau)$ on $t$ and $\tau$ varies from being relatively flat, near $\alpha = 1/2$, to relatively steep, near $\alpha = 3/2$.

Observe that fractional Brownian motion is *nonstationary*—its autocovariance is a function of "global" time $t$ as well as the time offset $\tau$. Most significantly, the variance of fBm is

$$\sigma^2(t) = \mathrm{E}\{|z(t)|^2\} = R_{zz}^{fBm}(t,0) = V_\alpha\,A^2|t|^{2\alpha-1} \tag{22}$$

which increases without bound; the longer one waits, the larger the expected amplitude of variability becomes. The time-varying fBm diffusivity is found to be

$$\kappa(t) = \frac{V_\alpha}{4}\frac{\alpha+1}{\alpha}\,A^2|t|^{2\alpha}, \qquad t \geq 0 \tag{23}$$

as we readily find by integrating the autocovariance as in (13). Like the variance, the diffusivity tends to increase without bound, rather than taking on a constant value. Note that the ratio of the diffusivity to the variance increases linearly with time, $\kappa(t)/\sigma^2(t) = 4|t|\alpha/(\alpha+1)$.

The normalizing constant in fBm, conventionally denoted $V_\alpha$, is defined as the variance at time $t=1$ of an fBm process having the amplitude parameter $A$ set to unity,

$$V_\alpha \equiv \mathrm{E}\{|z(1)|^2\} = R_{zz}^{fBm}(1,0), \qquad A = 1. \tag{24}$$

Its value is found to be (Barton and Poor, 1988)

$$V_\alpha = \frac{\Gamma(2-2\alpha)\sin(\pi\alpha)}{\pi(\alpha-1/2)} \tag{25}$$

where $\Gamma(x)$ is the gamma function. We find in Appendix B that this constant can be cast in the more symmetric form

$$V_\alpha = \frac{1}{\pi}\frac{\Gamma\left(\alpha-\frac{1}{2}\right)\Gamma\left(\frac{3}{2}-\alpha\right)}{\Gamma(2\alpha)} \tag{26}$$

which allows one to see behavior of this coefficient more clearly. Recall that $\Gamma(x)$, while positive for positive $x$, is negative in the interval $(-1,0)$, as follows from the reflection formula $\Gamma(x) = \pi/[\sin(\pi x)\Gamma(1-x)]$. Thus $V_\alpha$ is positive over the whole permitted range of $\alpha$, $1/2 < \alpha < 3/2$, but becomes unphysically negative as one passes outside of this range. Because the gamma function has a singularity at zero, with $\Gamma(x)$ tending to positive infinity as $x$ approaches zero from above, $V_\alpha$ also tends





to positive infinity as one approaches the two endpoints $\alpha = 1/2$ and $\alpha = 3/2$. Finally, from $\Gamma(1/2) = \sqrt{\pi}$ and $\Gamma(2) = 1$, the value of the coefficient for the Brownian case of $\alpha = 1$ is found to be $V_1 = 1$.

In addition to the autocovariance function, it is informative to also examine a related second-order statistical quantity,

$$\gamma_{zz}(t,\tau) \equiv \frac{1}{2}\mathrm{E}\left\{|z(t+\tau) - z(t)|^2\right\} = \frac{1}{2}\left[R_{zz}(t+\tau,0) + R_{zz}(t,0) - 2\Re\left\{R_{zz}(t,\tau)\right\}\right] \tag{27}$$

which is commonly known as the *variogram* in time series analysis and geostatistics, following Cressie (1988) and Matheron (1963); in the turbulence literature, the same quantity is widely used and is known as the *second-order structure function*, a term which dates back at least to the 1950's (Monin, 1958). For a stationary random process, the variogram becomes simply $\gamma_{zz}(t,\tau) = \gamma_{zz}(\tau) = \sigma^2 - \Re\{R_{zz}(\tau)\}$. Thus in the stationary case, the variogram merely repeats information already present in the autocovariance function.

For fractional Brownian motion, cancellations in the variogram occur and one obtains

$$\gamma_{zz}^{fBm}(t,\tau) = \gamma_{zz}^{fBm}(\tau) = \frac{V_\alpha}{2}A^2|\tau|^{2\alpha-1} \tag{28}$$

which is independent of global time $t$. Thus unlike its autocovariance function, the variogram of fBm is stationary. This equation states that the expected squared difference between fBm values at any two times is proportional to a power of the time difference, implying that the expected rate of growth of the fBm from its current value is independent of $t$. One might therefore

say that fBm is nonstationary, but in a time-independent or stationary manner. A process having a stationary variogram is said to be *intrinsically stationary* (Ma, 2004).

### 3.3    Linking the spectrum and autocovariance

Owing to its nonstationarity, the fBm autocovariance cannot be Fourier transformed in the usual way to yield a spectrum that is independent of global time $t$. Evidently the notion of what it means to be a Fourier transform pair must be generalized to

accommodate the time-dependent autocovariance. That the spectrum of fractional Brownian motion should be a power law of the form $|\omega|^{-2\alpha}$ was already conjectured by Mandelbrot and Van Ness (1968), based on earlier work by Hunt (1951) on the spectrum of its increments. Proving that this should be the case was accomplished by Solo (1992) using one approach, and by Flandrin (1989) and Øigård et al. (2006) using two variants of a different approach. Here, we essentially follow the latter paper, incorporating some additional details.

The Fourier transform with respect to $\tau$ of the nonstationary fBm autocovariance function $R_{zz}^{fBm}(t,\tau)$ will be found to be

$$S_{zz}^{fBm}(t,\omega) \equiv \int_{-\infty}^{\infty} R_{zz}^{fBm}(t,\tau)e^{-\mathrm{i}\omega\tau}\,\mathrm{d}\tau \tag{29}$$

$$= \frac{A^2}{|\omega|^{2\alpha}}\left(1 - e^{\mathrm{i}\omega t}\right) + V_\alpha A^2\pi|t|^{2\alpha-1}\delta(\omega) \tag{30}$$

where $\delta(t)$ is the Dirac delta function. This defines a time-varying version of the spectrum, described in more detail below, and was previously presented by Øigård et al. (2006), see their equation (8). Introducing a version of $S_{zz}^{fBm}(t,\omega)$ that is averaged

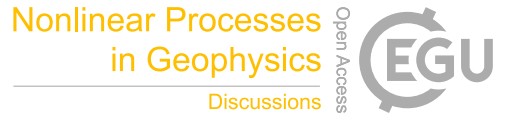

over a time interval $T$,

$$\overline{S_{zz}^{fBm}}(t,\omega;T) \equiv \frac{1}{T} \int\limits_{t-T/2}^{t+T/2} S_{zz}^{fBm}(u,\omega)\,\mathrm{d}u \tag{31}$$

where the faveraging time interval is chosen to be symmetrically located about the global time $t$, we find from (30)

$$\overline{S_{zz}^{fBm}}(t,\omega;T) = \frac{A^2}{|\omega|^{2\alpha}} \left[ 1 - e^{\mathrm{i}\omega t}\frac{\sin(\omega T/2)}{\omega T/2} \right] + \frac{V_\alpha}{2\alpha T}A^2\pi \left[ \left|t+\frac{T}{2}\right|^{2\alpha} - \left|t-\frac{T}{2}\right|^{2\alpha} \right]\delta(\omega). \tag{32}$$

5    One may then *define* the time-independent spectrum $S_{zz}^{fBm}(\omega)$ as the limit

$$S_{zz}^{fBm}(\omega) = \lim_{T\longrightarrow\infty} \overline{S_{zz}^{fBm}}(t,\omega;T) = \frac{A^2}{|\omega|^{2\alpha}} \tag{33}$$

where all terms dependent on global time $t$ are found to vanish.

This determines a sense in which the power-law form is the correct spectrum to associate with nonstationary fractional Brownian motion. In general, the Fourier transform with respect to $\tau$ of a nonstationary autocovariance function

$$\int\limits_{-\infty}^{\infty} R_{zz}(t,\tau)e^{-\mathrm{i}\omega\tau}\,\mathrm{d}\tau \equiv S_{zz}(t,\omega) \tag{34}$$

defines a time-varying generalization of the spectrum. The quantity $S_{zz}(t,\omega)$ is known as the Rihaczek (Rihaczek, 1968; Flandrin, 1999) or Kirkwood-Rihaczek (Kirkwood, 1933; Hindberg and Hanssen, 2007; Øigård et al., 2006) distribution, or alternatively by a more descriptive term, the *time-frequency spectral density* (Hanssen and Scharf, 2003). In the approach of Flandrin (1989), a different time-varying generalization of the spectrum is used instead of (30).

15    For fBm, there arises a complication to defining the time-frequency spectral density as in (34), because the integral in (29) is divergent. Despite this, (30) may be derived by interpreting this integral in a limiting sense, as is now shown. For $\alpha > 1/2$, consider the integral

$$\int\limits_{-\infty}^{\infty} |\tau|^{2\alpha-1}e^{-\mathrm{i}\omega\tau}\,\mathrm{d}\tau = 2\Re\left\{ \int\limits_{0}^{\infty} \tau^{2\alpha-1}e^{\mathrm{i}\omega\tau}\,\mathrm{d}\tau \right\} \tag{35}$$

which does not exist in the usual sense, since the integral is divergent. However, a limiting form does exist, given by

$$20 \quad \lim_{\epsilon\longrightarrow0} \int\limits_{0}^{\infty} \tau^{2\alpha-1}e^{-\epsilon\tau+\mathrm{i}\omega\tau}\,\mathrm{d}\tau = e^{\mathrm{i}\alpha\pi}\frac{\Gamma(2\alpha)}{\omega^{2\alpha}}, \qquad \omega \neq 0 \tag{36}$$

which is an example of what is termed an *Abel limit*, see Wong (1980, p. 407). Thus interpreting (35) as an Abel limit leads to

$$\frac{1}{2\cos(\pi\alpha)\Gamma(2\alpha)} \int\limits_{-\infty}^{\infty} |\tau|^{2\alpha-1}e^{-\mathrm{i}\omega\tau}\,\mathrm{d}\tau = \frac{1}{|\omega|^{2\alpha}} \tag{37}$$





such that a decaying power law in the frequency domain is associated with a growing power law, of one lower order, in the time domain. Here we have noted that changing the sign of $\omega$ in (36) is equivalent to a complex conjugation, since $(-1)^{2\alpha} = e^{2i\pi\alpha}$, this leading to the absolute value of $\omega$.

The coefficient of the integral in (37) simplifies to $-V_\alpha/2$, as shown in Appendix B. One then finds

$$-\int\limits_{-\infty}^{\infty} \frac{V_\alpha}{2} A^2 |\tau|^{2\alpha-1} e^{-i\omega\tau} \, d\tau = \frac{A^2}{|\omega|^{2\alpha}} = S_{zz}^{fBm}(\omega) \tag{38}$$

which shows that $A^2/|\omega|^{2\alpha}$ is the inverse Fourier transform, in the Abel limit sense, of that part of the nonstationary autocovariance function $R_{zz}^{fBm}(t,\tau)$ depending only on $\tau$. The inverse Fourier transformed quantity on the left-hand side of (38) is also recognized from (28) as the negative of the fBm variogram $\gamma_{zz}^{fBm}(\tau)$. A change of variables gives

$$-\int\limits_{-\infty}^{\infty} \frac{V_\alpha}{2} |t+\tau|^{2\alpha-1} e^{-i\omega\tau} \, d\tau = e^{i\omega t} \frac{1}{|\omega|^{2\alpha}} \tag{39}$$

and substituting (38) and (39) into (29), and making use of $\int_{-\infty}^{\infty} e^{-i\omega\tau} \, d\tau = 2\pi\delta(\omega)$, one obtains (30). From left to right in (30), we have the inverse Fourier transforms of the $|\tau|$ term, the $|t+\tau|$ term, and the $|t|$ term from the autocovariance.

This approach to proving that the power-law form is the correct spectrum to associate with fBm may be critiqued on the grounds that taking the limit of an average of the time-frequency spectral density, while mathematically sensible, does not correspond well with a limiting action that occurs in actual practice. Solo (1992) took a different approach, and found that if

the expected autocovariance and spectrum are estimated over a *finite* time interval, $S_{zz}^{fBm}(\omega) = A^2/|\omega|^{2\alpha}$ again emerges in the limit as that the time interval tends to infinity. That proof therefore has a strong intuitive appeal, but is more involved than the argument presented here.

### 3.4 Self-similarity

The most striking feature of fBm is that it is statistically identical to rescaled versions of itself. To show this, we define a time-

and amplitude-rescaled version of $z(t)$ as

$$\tilde{z}(t) \equiv \beta^{\alpha-1/2} z(t/\beta) \tag{40}$$

where the amplitude rescaling has been chosen to depend upon $\beta$ as well as the slope parameter $\alpha$. From (21), one finds

$$R_{\tilde{z}\tilde{z}}^{fBm}(t,\tau) = \beta^{2\alpha-1} R_{zz}^{fBm}(t/\beta, \tau/\beta) = \frac{V_\alpha}{2} A^2 \beta^{2\alpha-1} \left[ |(t+\tau)/\beta|^{2\alpha-1} + |t/\beta|^{2\alpha-1} - |\tau/\beta|^{2\alpha-1} \right]$$

$$= \frac{V_\alpha}{2} A^2 \left[ |t+\tau|^{2\alpha-1} + |t|^{2\alpha-1} - |\tau|^{2\alpha-1} \right] = R_{zz}^{fBm}(t,\tau) \tag{41}$$

and the autocovariance function of the rescaled process is determined to be the same as that of the original process.

Because the original process is Gaussian as well as zero mean, its statistical behavior is completely characterized by its autocovariance function. Thus fBm is statistically identical to itself when we "zoom in" in time, provided we also magnify





the amplitude appropriately. This property was referred to as *self-similarity* in the original work of Mandelbrot and Van Ness (1968); although later the term *self-affinity* was suggested as a substitute (Mandelbrot, 1985), the original term appears to be in more widespread use.

The positive constant $\beta$ can be seen as a temporal zoom factor, while the coefficient $\beta^{\alpha-1/2}$ describes how the amplitude is to be rescaled. Choosing $\beta > 1$ corresponds to zooming in in time, since then the interval from zero to $\beta$ in the new process $\tilde{z}(t)$ is drawn from the smaller interval zero to one in $z(t/\beta)$. Similarly, $\beta^{\alpha-1/2}$ with $\alpha > 1/2$ is greater than one, implying the amplitude must also be magnified. The required degree of amplitude magnification increases with $\alpha$ from a minimum value of unity at $\alpha = 1/2$ to a value of $\beta$ at $\alpha = 3/2$. The slope parameter $\alpha$ therefore governs the *aspect ratio* of rescaling for this self-similar behavior.

An illustration of self-similarity is presented in Fig. 5, using the real parts of the nine realizations shown in Fig. 4. The two panels show the effects of the self-similar rescaling (40) on each time series with a zoom factor $\beta = 4$, with the zooming represented by the gray boxes. The boxes on the left, of different aspect ratios, are rescaled according to the law (40) to have the same aspect ratios, as shown on the right. It is clear that each of the nine curves presents the same degree of roughness, and same amplitude of variability, on the left as on the right. This demonstrates what is meant by statistical self-similarity, and

shows how $\alpha$ controls the aspect ratio. A distinguishing feature of fractional Brownian motion is that this zooming may be continued indefinitely in either direction.

For stationary processes, self-similarity may also be seen in the frequency domain. Apply the rescaling (40) to some process $z(t)$, which is now assumed to be stationary. From the Fourier representation of the autocovariance, one finds

$$R_{\tilde{z}\tilde{z}}(\tau) = \beta^{2\alpha-1} \frac{1}{2\pi} \int_{-\infty}^{\infty} S_{zz}(\omega)\, e^{i\omega\tau/\beta} \mathrm{d}\omega = \beta^{2\alpha} \frac{1}{2\pi} \int_{-\infty}^{\infty} S_{zz}(\beta\omega)\, e^{i\omega\tau} \mathrm{d}\omega \tag{42}$$

after employing the change of variables $\omega/\beta \mapsto \omega$. Thus, in order for the process to be self-similar, one must have

$$S_{zz}(\omega) = \beta^{2\alpha} S_{zz}(\beta\omega) \tag{43}$$

in the spectral domain. This would clearly be the case for the power-law spectrum $S_{zz}(\omega) = A^2|\omega|^{-2\alpha}$, if a stationary process with such a spectrum were to exist. More generally, if a process has an *approximately* power-law spectrum over a range of frequencies, then the self-similarity condition (43) is expected to be *approximately* satisfied over that range.

Fractional Brownian motion is peculiar in that it has neither a well-defined derivative nor a well-defined integral. Loosely speaking, one may say that a derivative does not exist because the limiting action of taking a derivative conflicts with the self-similarity. Because $z(t)$ exhibits variability at infinitesimally small scales, $[z(t+\Delta) - z(t)]/\Delta$ does not have a well-defined limit as $\Delta$ tends to zero. The integral $\int_{-\infty}^{t} z(u)\, \mathrm{d}u$ does not exist either, because $z(t)$ has unbounded variance as $t$ progresses toward to infinitely large negative times and is therefore not integrable. Nevertheless, a *differenced* version of fBm does exist.

This process, termed *fractional Gaussian noise*, is discussed for completeness in Appendix C.



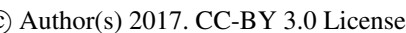



**Figure 5.** A demonstration of self-similarity for fractional Brownian motion, using the realizations presented in Fig. 4. The real part of each process is shown, with the $y$-axes in this figure corresponding exactly to the $x$-axis in Fig. 4. The gray boxes in panel (a) illustrate the different scaling behaviors, as described by (40). When each process is rescaled such that the boxes in (a) are transformed to the boxes in (b), the resulting time series are statistically identical to the originals. Thus the rescaled curves in (b) present the same degree of roughness as the corresponding curves in (a). The temporal magnification factor is $\beta = 4$, while the amplitude magnification factor $\beta^{\alpha-1/2}$ varies from 1 at $\alpha = 1/2$ to 4 at $\alpha = 3/2$. In order to avoid the appearance of additional roughness in (a) due only to numerical resolution, only every fourth point in (a) is shown; thus the curves in (a) and (b) consist of the same number of points.


### 3.5 Fractal dimension

The property of self-similarity, which is *global* in nature, was shown in the previous section to be related to the spectral slope. The slope is also related to two *local* properties, one associated with the slope at small frequencies, or the behavior of the autocovariance at large time offsets, and one associated with the slope at high frequencies, or the autocovariance at small time

offsets. The former property is the process *memory* or *long-range dependence* discussed in Section 2.4, while the latter is the *fractal dimension*.

Fractal dimension is a measure of the dimensionality of a curve (or some higher-order surface) that accounts the effect of roughness (Mandelbrot, 1985; Falconer, 1990). There are several different measures of fractal dimension in use, giving sometimes different values of dimensional measure for a particular curve (see e.g. Mandelbrot, 1985; Taylor and Taylor, 1991;

Dunbar et al., 1992). The most well-known measure, the Hausdorff dimension, is related to the behavior of the autocovariance function or variogram at very short time scales. One must also distinguish between the dimension of a curve as a function of the time variable, as in $u(t) = \Re\{z(t)\}$ versus $t$, and the dimension of a curve such as $z(t) = u(t) + \mathrm{i}v(t)$ in space or $u(t)$ versus $v(t)$, see e.g. Qian (2003). In the literature, the former is known as a *graph*, and the latter as a *sample path*.

The dimension of the graph is closely related to the short-time behavior of the autocovariance. As described by Gneiting

and Schlather (2004), for a univariate (or real-valued) stationary process $u(t)$ that has an autocovariance function behaving as $|\tau|^{\rho}$ for some $0 < \rho \leq 2$ as $\tau \to 0$, the Hausdorff dimension of the graph of the process is given by $D = 2 - \rho/2$. The comparable result for intrinsically stationary processes such as fBm is provided by Adler (1981). For fBm, $\rho = 2\alpha - 1$, hence the dimension of the graph of (real-valued) fBm is $D = 5/2 - \alpha$. This varies from $D = 1$ for the smoothest processes having $\alpha = 3/2$, to $D = 2$ for the roughest processes with $\alpha = 1/2$, corresponding to the bottom-to-top progression seen in Fig. 5.

As pointed out by Gneiting and Schlather (2004), the self-similarity of fBm links the behavior at very large scales and very small scales together. Because the spectral slope is constant, the fractal behavior at small scales implies a singularity at the origin. This is associated with unbounded diffusivity, and since the singularity is not integrable, with unbounded variance as well. The Matérn process examined in the next section has an additional degree of freedom with respect to fBm, such that the spectrum transitions to flat values for sufficiently low frequencies. This decouples the fractal dimension from the low-frequency

behavior and permits the phenomenon of diffusivity to arise.

### 3.6 Definition as a stochastic integral equation

Fractional Brownian motion is defined via the stochastic integral equation (Mandelbrot and Van Ness, 1968)

$$z(t) = \frac{A}{\Gamma(\alpha)} \left\{ \int_{-\infty}^{0} \left[ (t-s)^{\alpha-1} - (-s)^{\alpha-1} \right] \mathrm{d}W(s) + \int_{0}^{t} (t-s)^{\alpha-1} \mathrm{d}W(s) \right\} \tag{44}$$

where $\mathrm{d}W(t)$ here are increments of the complex-valued Wiener process, the covariance of which between itself at two different

times is

$$E\{\mathrm{d}W(t)\,\mathrm{d}W^*(s)\} = \delta(t-s)\,\mathrm{d}t\,\mathrm{d}s. \tag{45}$$





The integration with respect to $\mathrm{d}W(s)$ indicates in (44) that these integrals are of the Riemann-Stieltjes form, see Percival and Walden (1993). The process $\mathrm{d}W(t)$ can be said to represent continuous-time white noise, thus this equation defines fBm as a weighted integral of white noise. Further intuitive content of (44) is not initially apparent, so we will take some time to examine it in detail.

Note that standard Brownian motion, corresponding to $\alpha = 1$, is defined for all $t$ as

$$z(t) = A \int\limits_0^t \mathrm{d}W(s) \tag{46}$$

in which the integral is interpreted as $z(t) = -A \int_t^0 \mathrm{d}W(s)$ for $t < 0$. Intuitively, this is simply an integral of white noise. The fBm definition (44) reduces to the Brownian form with $\alpha = 1$, with the term on the first line of (44) vanishing.

The stochastic integral equation (44) can be written in the somewhat more transparent form

$$z(t) = \frac{1}{\Gamma(\alpha)} \int\limits_{-\infty}^t \left[ (t-s)^{\alpha-1} - I(-s)(-s)^{\alpha-1} \right] \mathrm{d}W(s) \tag{47}$$

where $I(t)$ is the indicator, or unit step, function defined as

$$I(t) \equiv \begin{cases} 1, & t \geq 0 \\ 0, & t < 0 \end{cases}. \tag{48}$$

Note that the two components of (47) cannot be written as separate integrals, because they are based on the same realization of $\mathrm{d}W(s)$. The purpose of the second term in (47) is now clearly seen to set the initial condition. It is not a function of time; it is

simply a random number, chosen to set $z(0) = 0$ identically.

The weighting factors such as $(t-s)^{\alpha-1}$ in (44) may be seen as creating a *fractional integral* of the Wiener process, as will now be shown. There is a simple expression for a function $f(t)$ that is integrated $n$ times from some initial point $a$ to time $t$, an action that is the reverse of the repeated derivative $(\mathrm{d}^n/\mathrm{d}t^n)f(t)$. This formula, known as *Cauchy's formula for repeated integration*, states

$$\int\limits_a^t \int\limits_a^{\tau_1} \cdots \int\limits_a^{\tau_{n-1}} f(\tau_n)\,\mathrm{d}\tau_n \cdots \mathrm{d}\tau_2\,\mathrm{d}\tau_1 = \frac{1}{(n-1)!} \int\limits_a^t (t-\tau)^{n-1} f(\tau)\,\mathrm{d}\tau \tag{49}$$

meaning that one may collapse an integral that is repeated $n$ times into a single integral, with a weighting to the $(n-1)$th power. Note that applying $(\mathrm{d}^n/\mathrm{d}t^n)$ to both sides, one obtains $f(t) = f(t)$—the left-hand side by repeated applications of the fundamental theorem of calculus, and the right-hand side by repeated applications of the Leibniz integral rule.

While the left-hand side of the Cauchy integral formula is not interpretable for non-integer $\alpha$, the right-hand side remains

valid. This allows us to *define* a fractional integral of $f(t)$ by letting $n$ take on non-integer values in the right-hand-side of (49). According to this reasoning, the quantity

$$\frac{1}{\Gamma(\alpha)} \int\limits_a^t (t-\tau)^{\alpha-1} f(\tau)\,\mathrm{d}\tau \qquad \alpha > 0 \tag{50}$$



is known as the Riemann-Liouville fractional integral, and may be said to integrate the function $f(t)$ a *fractional* number of times $\alpha$. For further details on fractional calculus, see e.g. Gorenflo and Mainardi (1997).

Returning to the definition of fBm in (44), we now see that it is simply a fractional integral of continuous-time white noise, modified to have the initial condition $z(0) = 0$. Unlike standard Brownian motion (46), which is integrated only from time
$t = 0$, for fractional Brownian motion one integrates from the infinite past in order to obtain the desired statistical behavior, and then one offsets this process by the correct amount in order to set the desired initial condition.

## 4    The Matérn process

The previous section reviewed the properties of fractional Brownian motion, including its self-similarity and fractal dimension, and showed how these are related to the spectral slope. This section examines the Matérn process in detail, with a focus on its
relationship to fBm. A simple extensions, the addition of a 'spin parameter', generalizes the Matérn process to encompass a larger family of oscillatory processes that are shown to represent forced/damped fractional oscillators.

### 4.1    The Matérn process and its spectrum

In Section 2 we showed that fractional Brownian motion is unable to capture long-time diffusive behavior, and demonstrated that this was a deficiency for the particular application to modeling particle velocities in two-dimensional turbulence. Regarding
the spectra in Fig. 3, one sees a high-frequency power law slope but a low-frequency plateau. This leads us to consider a spectrum of the form

$$S_{zz}^{M}(\omega) = \frac{A^2}{(\omega^2 + \lambda^2)^{\alpha}}, \qquad \alpha > \frac{1}{2} \tag{51}$$

which is the spectrum of a type of stationary random process known as the Matérn process (Matérn, 1960; Guttorp and Gneiting, 2006). Compared with fBm, the Matérn spectrum incorporates an additional (non-negative) parameter $\lambda$ having units
of frequency, which will be shown to have the physical interpretation of a damping. Note that the form of the Matérn spectrum also generalizes that of the Ornstein-Uhlenbeck process, which corresponds to the $\alpha = 1$ case, to fractional orders (Wolpert and Taqqu, 2005; Lim and Eab, 2006).

Examples of simulated Matérn processes are shown in Fig. 6, for twelve different values of $\alpha$ and three different values of $\lambda$. The box indicates a very low-damping regime with $1/2 < \alpha < 3/2$, roughly corresponding to the fractional Brownian motion
realizations seen in Fig. 4. There are two important differences when compared to fBm. The first is that there is no upper bound on $\alpha$, so processes can become still smoother than the $\alpha = 3/2$ case that defines the upper limit of the slope parameter for fBm. The second is the role of the additional parameter $\lambda$. As this parameter is increased, the curves for any $\alpha$ value appear more and more like white noise.



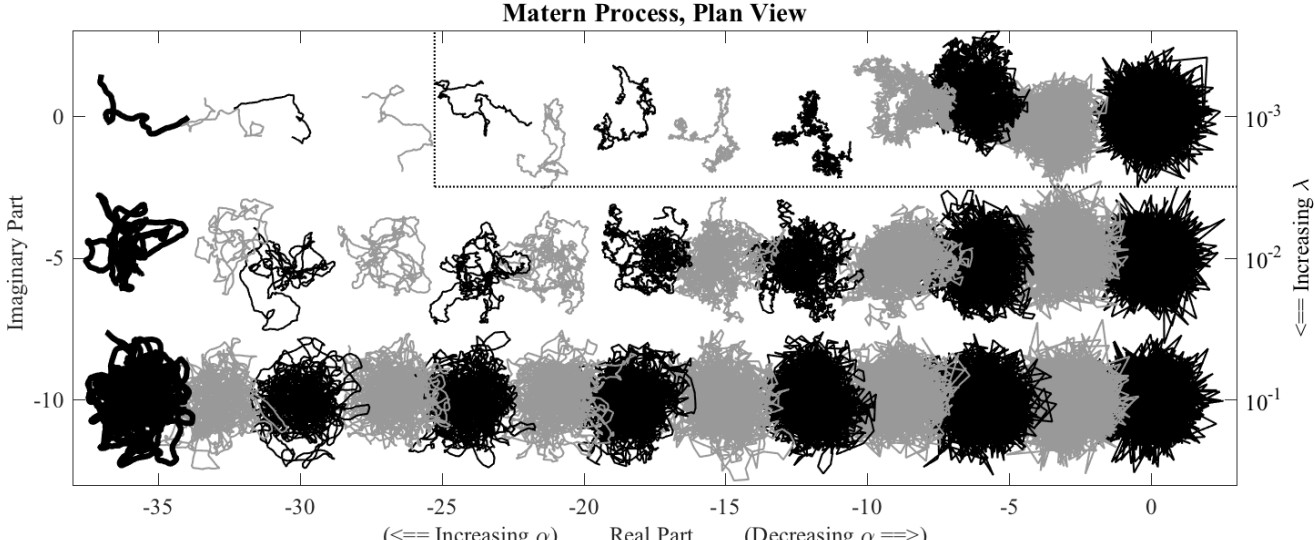

**Figure 6.** Plan view of realizations of the complex-valued Matérn process, for twelve different values of the slope parameter $\alpha$ and three different values of the damping parameter $\lambda$. Lines corresponding to successively higher values of $\alpha$ are offset by a value of $-3$ in the $x$-direction, while successively higher values of $\lambda$ are offset by a value of $-3$ in the $y$-direction. The slope parameter $\alpha$ ranges from just greater than 1/2 to 2 with an interval of 1/8, while $\lambda$ takes the values 1/10, 1/100, and 1/1000. The various $\alpha$ values are shown as alternating black and gray lines, with largest value $\alpha = 2$ shown as the heavy black line. The dotted box corresponds to those values of $\alpha$ shown previously in Fig. 4, for the smallest of the three damping values shown here.

The damping parameter $\lambda$ thus emerges as controlling the transition between two distinct spectral regimes. The Matérn spectrum is observed to have two limits

$$S^M_{zz}(\omega) \approx \frac{A^2}{|\omega|^{2\alpha}}, \qquad |\omega| \gg \lambda \tag{52}$$

$$S^M_{zz}(\omega) \approx \frac{A^2}{\lambda^{2\alpha}}, \qquad |\omega| \ll \lambda \tag{53}$$

so that, for high frequencies, an fBm-like power-law decay is recovered, while for low frequencies the spectrum approaches a constant and therefore may be said to be "locally white" for small $|\omega|/\lambda$. The Matérn process thus provides a continuum between the two regimes of white noise and a power-law spectrum, with a transition dictated by the value of $\lambda$.

The theoretical spectra corresponding to the realizations in Fig. 6 are shown in Fig. 7a. When frequency is normalized by the damping parameter, the theoretical (as opposed to the sampled) spectra for the different $\lambda$ values become identical. A transition

in the vicinity of $\omega/\lambda = 1$ is readily apparent. The different spectral levels reflect the choice of normalization, which is that $\sigma^2$ is has been set to unity. Smaller values of $\alpha$, corresponding to slower decay, therefore appear with lower spectral levels in order to integrate to unit variance.





To examine the role of $\lambda$ as a transition frequency, we take the derivative of the logarithm of the spectrum, and obtain

$$\frac{\mathrm{d}}{\mathrm{d}\omega} \ln S_{zz}^M(\omega) = -\alpha \frac{2\omega}{\omega^2 + \lambda^2} \tag{54}$$

the derivative which vanishes at $|\omega| = \lambda$. At this frequency the second derivative of (54) is negative. Thus the parameter $\lambda$ gives the frequency at which $\ln S_{zz}^M(\omega)$ is decreasing most rapidly with increasing $|\omega|$, which is a natural measure of the

transition point between the energetic "white" regime at low frequencies and the decaying regime at high frequencies. Since $\frac{d}{d\omega} \ln S_{zz}^M(\omega) = \left[\frac{d}{d\omega} S_{zz}^M(\omega)\right] / S_{zz}^M(\omega)$, $|\omega| = \lambda$ is the frequency at which the *fractional* decrease in $S_{zz}^M(\omega)$ is largest.

The variance and diffusivity of the Matérn process are both finite, and are found to be given by

$$\sigma^2 = c_\alpha \frac{A^2}{\lambda^{2\alpha-1}}, \qquad \kappa = \frac{1}{4} \frac{A^2}{\lambda^{2\alpha}} \tag{55}$$

in which we have introduced the normalizing constant

$$c_\alpha \equiv \frac{1}{2\pi} B\left(\frac{1}{2}, \alpha - \frac{1}{2}\right) = \frac{1}{2\pi} \frac{\Gamma\left(\frac{1}{2}\right) \Gamma\left(\alpha - \frac{1}{2}\right)}{\Gamma(\alpha)} \tag{56}$$

where $B(x,y) \equiv \Gamma(x)\Gamma(y)/\Gamma(x+y)$ is the beta function. The value of the diffusivity is found from $\kappa = S_{zz}(0)/4$, see (6), together with the Matérn spectrum (51), while the variance is

$$\sigma^2 = \frac{1}{2\pi} \int_{-\infty}^{\infty} S_{zz}^M(\omega)\, \mathrm{d}\omega = \frac{1}{2\pi} \int_{-\infty}^{\infty} \frac{A^2}{(\omega^2 + \lambda^2)^\alpha}\, \mathrm{d}\omega = \frac{A^2}{2\pi\lambda^{2\alpha-1}} \int_0^{\infty} \frac{x^{-1/2}}{(1+x)^\alpha}\, \mathrm{d}x \tag{57}$$

after the change of variables $\omega^2 \mapsto x\lambda^2$. Applying one of the defining forms of the beta function, Gradshteyn and Ryzhik (2000,

3.194.3),

$$\int_0^{\infty} \frac{x^{\mu-1}}{(1+x)^\nu}\, \mathrm{d}x = B(\mu, \nu - \mu), \qquad \nu > \mu > 0 \tag{58}$$

then leads to the variance expression given in (55).

The Matérn spectrum can then be rewritten in terms of the variance $\sigma^2$ as

$$S_{zz}^M(\omega) = \frac{\lambda^{2\alpha-1}}{c_\alpha} \frac{\sigma^2}{(\omega^2 + \lambda^2)^\alpha} \tag{59}$$

so that the diffusivity becomes $\kappa = \frac{1}{4}\sigma^2/(\lambda c_\alpha)$. In this form, the Matérn spectrum becomes a function of $\sigma^2$, $\alpha$, and $\lambda$ rather than $A^2$, $\alpha$, and $\lambda$. This will prove to be more convenient for numerical optimization during parametric fitting, because reasonable ranges for $\sigma$ are more readily determined than are ranges of $A$. This re-parameterization also simplifies somewhat the form of the autocovariance function, presented next.

### 4.2 The autocovariance function

The autocovariance function corresponding to the spectrum (51) is found to be (Matérn, 1960; Guttorp and Gneiting, 2006)

$$R_{zz}^M(\tau) = \frac{2\sigma^2}{\Gamma(\alpha - 1/2)\, 2^{\alpha-1/2}} |\lambda\tau|^{\alpha-1/2} \mathcal{K}_{\alpha-1/2}(\lambda|\tau|) \tag{60}$$





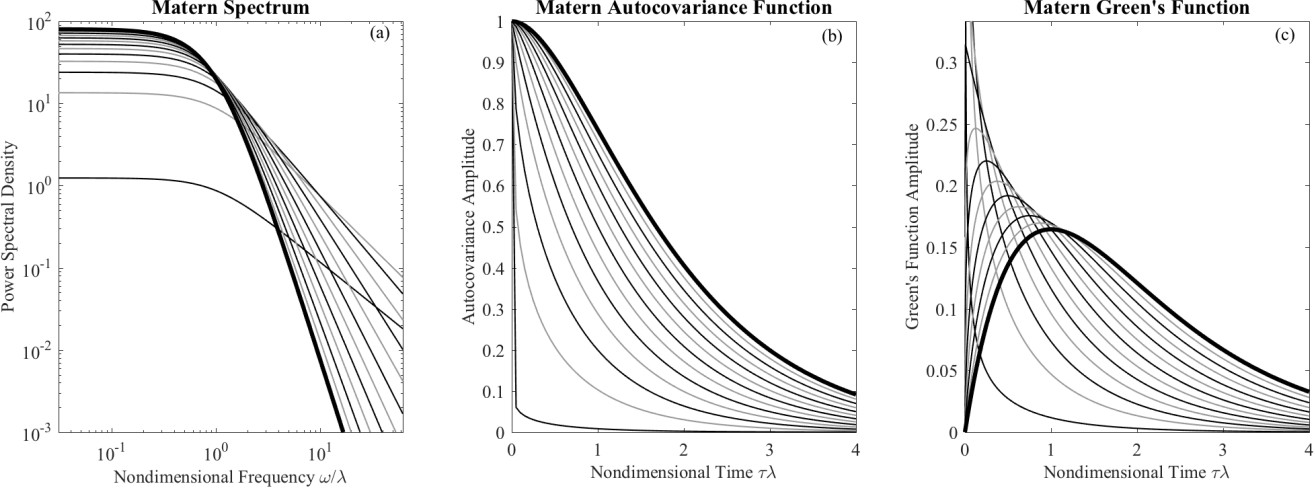

**Figure 7.** Theoretical spectra (a), autocovariance functions (b), and Green's functions (c) for Matérn processes corresponding to the different $\alpha$ values shown previously in Fig. 6, and with the process variance set to $\sigma^2 = 1$. The corresponding expressions are (59), (60), and (72), respectively. As in Fig. 6, the various $\alpha$ values are shown by alternating black and gray lines, with $\alpha = 2$ shown as a heavy black line. Time and frequency have been nondimensionalized as $\tau\lambda$ and $\omega/\lambda$, respectively; thus the transition between a flat and a sloped regime occurs in the vicinity of $\omega/\lambda = 1$ in (a), while the $e$-folding time in (c) is $\tau\lambda = 1$. The autocovariance function (b) develops a strong singularity as $\alpha$ approach 1/2, which is linked to the flattening of the spectrum in (a). The Green's function in (c) is infinite at $\tau\lambda = 0$ for $\alpha < 1$, and vanishes at $\tau\lambda = 0$ for $\alpha > 1$.

where $\mathcal{K}_\nu(x)$ is the modified Bessel function of the second kind of order $\nu$. For $|\tau| \gg 1/\lambda$, one has the behavior

$$R_{zz}^M(\tau) \approx \frac{\sqrt{2\pi}\,\sigma^2}{\Gamma(\alpha - 1/2)\,2^{\alpha - 1/2}}\, |\lambda\tau|^{\alpha - 1}\, e^{-\lambda|\tau|} \tag{61}$$

as follows from the asymptotic behavior of the modified Bessel function for large argument Abramowitz and Stegun (1972, 9.7.2). Thus the Matérn process exhibits exponential decay of its covariance function, and is therefore categorized as a *short-*

5 *memory* process.

Examples of theoretical Matérn autocovariance functions are presented in Fig. 7b, again corresponding to the realizations in Fig. 6. As is usual with Fourier pairs, the most localized spectra correspond to the most distributed autocovariance functions, and vice-versa. As $\alpha$ decreases, the autocovariance falls off more and more quickly from the origin, with a singularity developing at the origin as $\alpha$ approaches one-half.

10 For time offsets that are small compared to the damping timescale, $|\tau| \ll 1/\lambda$, and for the slope parameter in the range $1/2 < \alpha < 3/2$, one finds

$$R_{zz}^M(\tau) \approx \sigma^2 \left[ 1 - \left(\frac{\lambda|\tau|}{2}\right)^{2\alpha - 1} \frac{\Gamma\left(\frac{3}{2} - \alpha\right)}{\Gamma\left(\alpha + \frac{1}{2}\right)} \right] \tag{62}$$

as the short-time behavior of the Matérn autocovariance function. This is derived in Appendix D following Goff and Jordan (1988, their Appendix A), who were apparently the first to establish it, see Guttorp and Gneiting (2006). It is also shown in





Appendix D that for $\alpha > 3/2$, the lowest-order dependence of the Matérn autocovariance function no longer contains a power of $\alpha$, but instead remains proportional to $\tau^2$ as $\alpha$ increases.

The expression (62) for the short-time behavior of the Matérn autocovariance may be simplified by noting

$$c_\alpha \frac{1}{2^{2\alpha-1}} \frac{\Gamma\left(\frac{3}{2} - \alpha\right)}{\Gamma\left(\alpha + \frac{1}{2}\right)} = \frac{1}{2} V_\alpha \qquad (63)$$

which relates $V_\alpha$, the coefficient of fractional Brownian motion defined in (26), to $c_\alpha$, the normalizing constant for the Matérn process defined in (56). These two definitions together with the duplication for the gamma function (B5) presented in Appendix B lead to the above result. Substituting this into the asymptotic expansion (62) for small $|\tau|$, we obtain for $1/2 < \alpha < 3/2$

$$R_{zz}^M(\tau) \approx \sigma^2 - \frac{1}{2} V_\alpha A^2 |\tau|^{2\alpha-1}, \qquad |\tau| \ll 1/\lambda \qquad (64)$$

after making use of the expression for the Matérn variance given by (55). This matches exactly the $\tau$-dependence inferred for a power-law spectrum using the Abel limit (38). Note that the only dependence on $\lambda$ of the autocovariance for $|\tau| \ll 1/\lambda$ is through the variance $\sigma^2$.

From this small-$\tau$ expansion, we can immediately determine the fractal dimension, as discussed in Section 3.5. One finds

$$D = \begin{cases} \frac{5}{2} - \alpha & \alpha < 3/2 \\ 1 & \alpha \geq 3/2 \end{cases} \qquad (65)$$

so that the fractal dimension decays from $D = 2$, for very rough processes with $\alpha = 1/2$, to $D = 1$, for smooth processes with $\alpha = 3/2$, just as with fractional Brownian motion. For slopes steeper than $\omega^{-3}$, the fractal dimension remains at unity. This is a consequence of the fact that the highest power of $\tau$ appearing in the small-$\tau$ expansion (62) is $\tau^2$.

Next we verify that the Matérn spectrum (59) and autocovariance function (60) are indeed a Fourier transform pair. The integral relation 17.34.9 given on p. 1126 of Gradshteyn and Ryzhik (2000) is

$$\frac{\sqrt{2}}{\Gamma(\nu + 1/2)} \left(\frac{|\tau|}{2\lambda}\right)^\nu K_\nu(\lambda|\tau|) = \sqrt{\frac{2}{\pi}} \int\limits_0^\infty \frac{\cos(\omega\tau)}{(\omega^2 + \lambda^2)^{\nu+1/2}} \, \mathrm{d}\omega \qquad (66)$$

for $\nu > -1/2$ and $\lambda > 0$. This is sometimes known as Basset's formula, see Watson (1922, p. 172), who states that this result for the case of integer $\nu$ is originally due to Basset (1888, p. 19), and who also discusses some history of the integral on the right-hand-side. A rearrangement shows that (60) is the inverse Fourier transform of (59).

## 4.3   Addition of spin

A very simple modification can expand the range of possibilities of the Matérn process, and also aid in the development of physical intuition. We add a deterministic tendency for the process to spin on the complex plane at rate $\Omega$, and refer to this new





process as the oscillatory Matérn process or oMp. Modulating the Matérn autocovariance $R_{zz}^M(\tau)$ by $e^{i\Omega\tau}$ gives

$$R_{zz}^{oMp}(\tau) = e^{i\Omega\tau} R_{zz}^M(\tau) \tag{67}$$

$$S_{zz}^{oMp}(\omega) = \frac{A^2}{[(\omega - \Omega)^2 + \lambda^2]^\alpha} \tag{68}$$

for the new autocovariance function / spectrum pair. Note that with $\alpha = 1$, these reduce to

$$R_{zz}^{oMp}(\tau) = \frac{A^2}{2\lambda} e^{i\Omega\tau} e^{-\lambda|\tau|} \tag{69}$$

$$S_{zz}^{oMp}(\omega) = \frac{A^2}{(\omega - \Omega)^2 + \lambda^2} \tag{70}$$

where we have made use of 10.2.17 on p. 444 of Abramowitz and Stegun (1972) for the former equality. These are observed to be the autocovariance and spectrum of the complex-valued oscillator known as the complex Ornstein-Uhlenbeck process (Jeffreys, 1942; Arató et al., 1999).

Thus the oscillatory Matérn process subsumes the Matérn process and the complex Ornstein-Uhlenbeck process into a larger family. In this next section we will determine the stochastic integral equation of this oscillatory Matérn process.

### 4.4 Stochastic integral equation

Unlike fractional Brownian motion, the Matérn process is not generally defined in terms of a stochastic integral equation or a stochastic differential equation. In this section, we will show that a stochastic integral equation that will generate an oscillatory Matérn process is

$$z(t) = A \int_{-\infty}^{t} g(t - s)\, dW(s) \tag{71}$$

where the Green's function, or impulse response function, is

$$g(t) \equiv \begin{cases} \frac{1}{\Gamma(\alpha)} t^{\alpha-1} e^{i\Omega t} e^{-\lambda t}, & t \geq 0 \\ 0, & t < 0 \end{cases}. \tag{72}$$

Note that the Green's function has been set to vanish before time $t = 0$, thus corresponding to a causal filter. The Fourier transform of a Green's function $g(t)$ is an important quantity known as the *transfer function*, and we find

$$G(\omega) = \int_{-\infty}^{\infty} g(t) e^{i\omega t} dt = \frac{1}{[i(\omega - \Omega) + \lambda]^\alpha} \tag{73}$$

for the Matérn transfer function, using 3.2.3 on p. 118 of Bateman (1954).

Examples of the Green's functions for $\Omega = 0$ are shown in Fig. 7c. Note a change in behavior across $\alpha = 1$. For higher values of $\alpha$, the Green's function vanishes at $\tau = 0$, thus developing a maximum that is seen to shift away from the origin as one increases $\alpha$. For $\alpha < 1$, however, a singularity develops at the origin, and the Green's function monotonically decays with increasing time.





Identifying this stochastic integral equation sheds light on the nature of the Matérn process itself. The Green's function $g(t)$ defined in (72) is also the solution to an impulse forcing of the damped fractional oscillator equation

$$\left[\frac{\mathrm{d}}{\mathrm{d}t} + \lambda - \mathrm{i}\Omega\right]^{\alpha} g(t) = \delta(t) \tag{74}$$

as will now be shown. We expand the fractional operator as

$$\left[\frac{\mathrm{d}}{\mathrm{d}t} + \lambda - \mathrm{i}\Omega\right]^{\alpha} = \sum_{n=0}^{\infty} \frac{\alpha(\alpha-1)\cdots(\alpha-n+1)}{n!}\left[\frac{\mathrm{d}^n}{\mathrm{d}t^n} + (\lambda - \mathrm{i}\Omega)^{\alpha-n}\right] \tag{75}$$

using Newton's generalization of the binomial theorem to non-integral orders (see e.g. Berge, 1971). Substituting $g(t) = \frac{1}{2\pi}\int_{-\infty}^{\infty}G(\omega)e^{\mathrm{i}\omega t}\mathrm{d}\omega$ into the left-hand side of (74), applying (75), and carrying out the indicated derivatives, leads to

$$\left[\frac{\mathrm{d}}{\mathrm{d}t} + \lambda - \mathrm{i}\Omega\right]^{\alpha} g(t) = \frac{1}{2\pi}\int_{-\infty}^{\infty}G(\omega)\left[\mathrm{i}\left(\omega - \Omega\right) + \lambda\right]^{\alpha} e^{\mathrm{i}\omega t}\mathrm{d}\omega \tag{76}$$

after collapsing the summation using a second application of the generalized binomial theorem. Now a cancellation occurs, and the right-hand side of (76) becomes simply $\frac{1}{2\pi}\int_{-\infty}^{\infty}e^{\mathrm{i}\omega t}\mathrm{d}\omega$, which is equal to $\delta(t)$, thus verifying (74).

This establishes the physical interpretation of the oscillatory Matérn process as a *damped fractional oscillator* forced by continuous-time white noise. The standard Matérn process is then seen as a forced/damped fractional oscillator in which the oscillation frequency is set to zero. As shown in the next section, fBm emerges if the damping also vanishes, with a modification in order to set an appropriate initial condition.

Now we show that the stochastic integral equation (71) using the Green's function (72) does indeed generate a random process having the Matérn spectrum. Note that because $g(t)$ vanishes for $t < 0$, we can write (71) with infinity as the upper limit of integration. The associated autocovariance function is

$$R_{zz}(\tau) \equiv \mathrm{E}\{z(t)z^*(t-\tau)\} = A^2 \int_{-\infty}^{\infty}\int_{-\infty}^{\infty}g(t-s)g^*(t-\tau-r)\mathrm{E}\{\mathrm{d}W(s)\mathrm{d}W^*(r)\} = A^2 \int_{-\infty}^{\infty}g(s)g^*(s-\tau)\mathrm{d}s \tag{77}$$

with the last expression following from the orthogonality property of the Wiener increments (45), together with a change in the variable of integration.

Expressing $g(t)$ in terms of its transfer function leads to the familiar cross-correlation theorem

$$\int_{-\infty}^{\infty}g(s)g^*(s-\tau)\mathrm{d}s = \frac{1}{2\pi}\int_{-\infty}^{\infty}|G(\omega)|^2 e^{\mathrm{i}\omega\tau}\mathrm{d}\omega \tag{78}$$

after making use of the Fourier representation of a delta function, $\delta(\omega) = \frac{1}{2\pi}\int_{-\infty}^{\infty}e^{\mathrm{i}\omega t}\mathrm{d}t$. From the form of the transfer function (73), we find the autocovariance function becomes

$$R_{zz}(\tau) = \frac{1}{2\pi}\int_{-\infty}^{\infty}\frac{A^2}{[(\omega-\Omega)^2 + \lambda^2]^{\alpha}}e^{\mathrm{i}\omega\tau}\mathrm{d}\omega \tag{79}$$





and therefore the spectrum of the process generated using the Green's function (72) matches the form for the oscillatory Matérn process (68), as claimed. As an aside, we point out that this result implies that with $\Omega = 0$, the cross-correlation of $g(t)$ with itself as in (77) must recover the Bessel function form of the Matérn autocovariance function, although this is not at all obvious in the time domain.

In the above discussion, we have avoided writing Matérn process as a stochastic *differential* equation, as there are mathematical difficulties in ensuring that the derivatives exist. The expansion of the fractional-order operator (75) involves infinitely many higher-order derivatives; but their existence conflicts with self-similar roughness of the Matérn process as one proceeds to increasingly small scales. The approach we have taken is intended to determine the physical nature of the system while sidestepping such mathematical difficulties.

## 10    4.5    Relationship to fractional Brownian motion

Having identified the stochastic integral equation for the Matérn process, we now examine its relationship with fractional Brownian motion. The Green's function of the oscillatory Matérn process (72) can be rewritten as

$$g_{\alpha,\lambda,\Omega}(t) \equiv \frac{1}{\Gamma(\alpha)} I(t) \, t^{\alpha-1} e^{i\Omega t} e^{-\lambda t} \tag{80}$$

where $I(t)$ is the indicator function defined in (48), and where we have explicitly specified the dependence of $g(t)$ upon the
Matérn parameters. In terms of this Green's function, the stochastic integral equation defining fBm (47) becomes

$$z(t) = \frac{A}{\Gamma(\alpha)} \int_{-\infty}^{t} \left[ (t-s)^{\alpha-1} - I(-s)(-s)^{\alpha-1} \right] dW(s) \tag{81}$$

$$= A \int_{-\infty}^{t} \left[ g_{\alpha,0,0}(t-s) - g_{\alpha,0,0}(-s) \right] dW(s). \tag{82}$$

The only difference between this and the equation for the Matérn process (71) is the second term in the integral, which as stated earlier, serves the function of enforcing the initial condition $z(0) = 0$. This confirms that the Matérn process is rightly thought
of as *damped* fractional Brownian motion.

If fractional Brownian motion and the standard Matérn processes are essentially facets of the same process, one should be able to see this directly from their autocovariances. This is indeed the case. For time shifts $\tau$ that are very small compared to the global time $t$, the fBm autocovariance (21) is approximately given by

$$R_{zz}^{fBm}(t,\tau) \approx \sigma^2(t) - \frac{1}{2} V_\alpha A^2 |\tau|^{2\alpha-1}, \qquad |\tau| \ll |t| \tag{83}$$

where $\sigma^2(t) \equiv R_{zz}^{fBm}(t,0) = V_\alpha A^2 |t|^{2\alpha-1}$ is the time-varying fBm variance encountered earlier in (22). This matches (64) the form of the Matérn autocovariance for small $|\tau|/\lambda$.

The intuitive interpretation of this result is that a Matérn process has a second-order structure that behaves for small time offsets $\tau$ in the same way as does fractional Brownian motion, considered for offsets $\tau$ that are small compared with the current



global time $t$. Or, even more succinctly, the *local* behaviors of the Matérn process and fBm are the same; they differ from each other only for sufficiently large time offsets.

To look at this another way, imagine that a modified Matérn process were constructed with an integral matching the form of that for fractional Brownian motion (82). In other words, we define $z(t)$ as in (82) but for arbitrary values of $\lambda$. Such a process

would then by definition have $z(0) = 0$, and would therefore not be stationary. For nonzero $\lambda$, after a sufficiently long time this initial condition is 'forgotten' on account of the decaying exponential in the Green's function, and the process will eventually behave as if it were stationary. For $\lambda = 0$, however, this initial condition is never forgotten.

## 5   Generation

This section addresses means to simulate realizations of fractional Brownian motion and the Matérn process numerically. The

main contribution is a new approach to simulate a diffusive process such as the Matérn in $O(N \log N)$ operations, by relying on the knowledge of its Green's function.

### 5.1   The Cholesky decomposition

The standard approach to simulating a Gaussian random process with a known covariance matrix is a method called the Cholesky decomposition, which we discuss here. In this section, as we will be dealing with vectors and matrices, a change of

notation is called for. We now let $z_n \equiv z(n\Delta t)$ with integer $n$ denote a discretely sampled random process, sampled at $N$ times separated by the uniform interval $\Delta$.

This sequence is arranged into a length $N$ random vector denoted $\mathbf{z}$. We define the expected $N \times N$ covariance matrix of $\mathbf{z}$ as $\mathbf{R} \equiv \mathrm{E}\left\{\mathbf{z}\mathbf{z}^H\right\}$, having components

$$R_{m,n} = \mathrm{E}\left\{z_m z_n^*\right\} = \mathrm{E}\left\{z\left(m\Delta t\right) z^*(n\Delta t)\right\} = R_{zz}\left(n\Delta t, (m-n)\Delta t\right). \tag{84}$$

Here $n\Delta t$ plays the role of global time $t$, and $(m-n)\Delta t$ that of the time offset $\tau$, in the evaluation of the nonstationary covariance function $R_{zz}(t,\tau) = \mathrm{E}\left\{z(t+\tau) z^*(t)\right\}$. Thus variation in $\mathbf{R}$ of the time offset $\tau$ with fixed global time $t$ occurs in the direction perpendicular to the main diagonal, while variation of $t$ with fixed $\tau$ occurs along the main diagonal. In the case of a stationary process, there is no variation parallel to the main diagonal, and $\mathbf{R}$ is then said to be a Toeplitz matrix.

The Cholesky decomposition factorizes the covariance matrix as $\mathbf{R} = \mathbf{L}\mathbf{U}$, where $\mathbf{L}$ is lower triangular and $\mathbf{U}$ is upper

triangular. It follows from the Hermitian symmetry of $\mathbf{R}$ that $\mathbf{L} = \mathbf{U}^H$ where the superscript "$H$" denotes the conjugate transpose. Now let $\mathbf{w}$ be an $N$-vector of unit-variance, independent, complex-valued Gaussian random variables. Forming the sequence $\hat{\mathbf{z}} = \mathbf{L}\mathbf{w}$, we find the covariance matrix $\widehat{\mathbf{R}} \equiv \mathrm{E}\left\{\hat{\mathbf{z}}\hat{\mathbf{z}}^H\right\}$ associated with $\hat{\mathbf{z}}$ is given by

$$\widehat{\mathbf{R}} \equiv \mathbf{L}\,\mathrm{E}\left\{\mathbf{w}\mathbf{w}^H\right\}\mathbf{L}^H = \mathbf{L}\mathbf{I}\mathbf{L}^H = \mathbf{R} \tag{85}$$

where $\mathbf{I}$ is the $N \times N$ identity matrix. Note while we could have also chosen to use $\mathbf{U}$ to generate the random sequence, the

use of $\mathbf{L}$ is preferable as it corresponds to a causal filter.





Thus to simulate a length $N$ sequence of a possibly nonstationary Gaussian random process, one simply populates an $N \times N$ matrix with the known values from the autocovariance function, applies the Cholesky decomposition, and multiplies the result by a vector of white noise. The resulting sequence has the *identical* covariance structure to a length $N$ sample of the corresponding random process.

A limitation of this approach is that the Cholesky decomposition requires $O(N^3)$ operations. Computational costs therefore increase steeply. However, it is the case that many realizations of sequences of a fixed length can be generated quickly, because one only needs to form the Cholesky decomposition once for a given autocovariance matrix. For simulation of stationary processes, the Toeplitz matrix structure can in principle be used to accelerate the Cholesky decomposition to $O(N^2)$ or even $O(N \log N)$, see Yagle and Levy (1985) and Dietrich and Newsam (1997) respectively. The latter method, termed circulant

embedding, while $O(N \log N)$, involves embedding the covariance matrix of interest within a larger matrix, and leads somewhat unpredictable tradeoff between minimizing error and increasing the matrix size (Percival, 2006). The method presented here has the advantages that it is very straightforward to implement, and that the error terms are well understood provided the Green's function is known.

## 5.2 Discretization effects in fast generation

To devise our generation method, we will employ a finer temporal spacing $\tilde{\Delta} \equiv \Delta/k$, where $k$ is a positive integer termed the *oversampling parameter*. The Green's function integral for the Matérn process (71) can be rewritten as the sum of smaller integrals over segments of duration $\tilde{\Delta}$

$$z(t) = A \sum_{p=0}^{\infty} \int_{t-(p+1)\tilde{\Delta}}^{t-p\tilde{\Delta}} g(t-s)\, dW(s) \tag{86}$$

which remains exact. Now for each of these integrals over a short segment, we approximate the Green's function by a constant,

namely the value of the Green's function at the segment midpoint, which occurs when $t - s = (p+1/2)\tilde{\Delta}$. Employing this approximation and evaluating the result at the discrete times $t = n\Delta$ *defines* a discrete series

$$\tilde{z}_n = A \sum_{p=0}^{\infty} g\left((p+1/2)\tilde{\Delta}\right) \int_{n\Delta-(p+1)\tilde{\Delta}}^{n\Delta-p\tilde{\Delta}} dW(s) \tag{87}$$

for all integers $n = -\infty, \ldots, -2, -1, 0, 1, 2, \ldots \infty$. Because $\int_a^b dW(s)$ is a zero-mean Gaussian random variable with variance $(b-a)$, the integral in the above expression simplifies to

$$\int_{n\Delta-(p+1)\tilde{\Delta}}^{n\Delta-p\tilde{\Delta}} dW(s) = \int_{(nk-p-1)\tilde{\Delta}}^{(nk-p)\tilde{\Delta}} dW(s) = w_{nk-p}\sqrt{\tilde{\Delta}} \tag{88}$$

where $w_n$ is a sequence of complex-valued, unit variance, independent Gaussian random variables. Finally, introducing an oversampled version of the discrete Green's function as

$$g_n^{\{k\}} = g((n+1/2)\Delta/k) \tag{89}$$





our expression (87) for $\tilde{z}_n$ becomes

$$\tilde{z}_n = A\frac{\Delta}{k}\sum_{p=0}^{\infty} g_p^{\{k\}} w_{nk-p}. \tag{90}$$

This is a discrete convolution, but modified by the fact that the output will have a temporal resolution that is $k$ times more coarse than that of the two input series.

The autocovariance function of $\tilde{z}_n$ is very close to the sampled autocovariance function of the Matérn process, and can be made arbitrary close by a suitable choice of oversampling rate $k$, as will now be shown. The autocovariance sequence associated with $\tilde{z}_n$ is found to be

$$\widetilde{R}_n \equiv \mathrm{E}\left\{\tilde{z}_m \tilde{z}_{m-n}^*\right\} = A^2\frac{\Delta^2}{k^2} \times \sum_{p=0}^{\infty}\sum_{q=0}^{\infty} g_p^{\{k\}}\left[g_q^{\{k\}}\right]^* \mathrm{E}\left\{w_{mk-p}\, w_{(m-n)k-q}^*\right\} \tag{91}$$

and since $\mathrm{E}\{w_m w_n^*\} = \delta_{m,n}$ where $\delta_{m,n}$ is the Kronecker delta function, all terms in the summation vanish except when
$mk - p = (m-n)k - q$ or equivalently $q = p - nk$. Thus

$$\widetilde{R}_n = \mathrm{E}\left\{\tilde{z}_m \tilde{z}_{m-n}^*\right\} = A^2\frac{\Delta^2}{k^2}\sum_{p=0}^{\infty} g_p^{\{k\}}\left[g_{p-nk}^{\{k\}}\right]^* \tag{92}$$

which is clearly an approximation to (77) for an autocovariance function in terms of its Green's function. The discretely sampled autocovariance sequence can therefore be approximated to arbitrary precision by a choosing a suitable degree of oversampling. However, notice that the summations in (90) and (92) extend to infinity, which is not possible in practice. In the
next subsection we examine the impact of additional errors resulting from finite sample size effects.

### 5.3    Sample size effects in fast generation

In practice, the summations over the duration of the Green's function must be truncated at some point. It is tempting to truncate the Green's function after a relatively short time. However, for spectra having a large dynamic range, this truncation leads to undesirable leakage effects, just as in spectral analysis. Instead, we will utilize the entire length of the time series. Anticipating
transforming to the Fourier domain, we define sequences to be periodized.

Firstly we need to determine a suitable cutoff for limiting the long-term influence of the Green's function. We denote by $T_\epsilon$ the time such that the magnitude of the Green's function, integrated to this time, rises within $\epsilon$ of the value it obtains when integrated over all times:

$$\frac{\int_0^{T_\epsilon} |g(s)|\, ds}{\int_0^{\infty} |g(s)|\, ds} = 1 - \epsilon. \tag{93}$$

Using the definition of the Matérn Green's function (72), one may readily show that this occurs when

$$\frac{\gamma(\alpha, \lambda T_\epsilon)}{\Gamma(\alpha)} = 1 - \epsilon, \qquad \gamma(\alpha, t) \equiv \int_0^t s^{\alpha-1} e^{-s}\, ds. \tag{94}$$



where $\gamma(\alpha, t)$ is the incomplete gamma function of order $\alpha$ evaluated at time $t$.

We assume that the length of the series we will be generating, $T = N\Delta$, is much larger than the cutoff time for some chosen value of epsilon, i.e. $T \gg T_\epsilon$. Because we intend to employ a periodic convolution, yet wish to prevent noise values at the end of the time series from influencing the beginning, we will create a sequence of length $\widehat{N} \equiv N + N_\epsilon$. Let $\widehat{w}_n$ be a version of the noise that is periodic with period $\widehat{N}$, and $\widehat{g}_n^{\{k\}}$ be a version of $g_n^{\{k\}}$ that is set to zero for $n > \widehat{N} - 1$. Form a length-$\widehat{N}$ vector $\hat{\mathbf{z}}$ with entries given by

$$\hat{z}_n \equiv A \frac{\Delta}{k} \sum_{p=0}^{\widehat{N}-1} \widehat{g}_p^{\{k\}} \widehat{w}_{nk-p} \tag{95}$$

and now decompose this vector into two parts, $\hat{\mathbf{z}} = [\hat{\mathbf{z}}_\epsilon \; \hat{\mathbf{z}}_o]^T$ where the superscript "$T$" is the transpose operator. In the initial portion $\hat{\mathbf{z}}_\epsilon$, of length $N_\epsilon$, the decaying Green's function is interacting with periodic noise from the end of the time series. This portion is discarded, while the second portion $\hat{\mathbf{z}}_o$ is of length $N$ and is the simulated series we desire.

The $N \times N$ covariance matrix associated with the latter sequence, $\widehat{\mathbf{R}} = \mathrm{E}\{\hat{\mathbf{z}}_o \hat{\mathbf{z}}_o^H\}$, has components given by

$$\widehat{R}_{m,n} = A^2 \frac{\Delta^2}{k^2} \sum_{p=0}^{\widehat{N}-1} \sum_{q=0}^{\widehat{N}-1} \widehat{g}_p^{\{k\}} \left[\widehat{g}_q^{\{k\}}\right]^* \mathrm{E}\left\{\widehat{w}_{mk-p+kN_\epsilon} \widehat{w}_{nk-q+kN_\epsilon}^*\right\}. \tag{96}$$

To simplify this expression, observe that the covariance of the periodized noise sequence $\widehat{w}_n$ is

$$\mathrm{E}\{\widehat{w}_m \widehat{w}_n^*\} = \sum_{\ell=-\infty}^{\infty} \delta_{m,n+\ell\widehat{N}} \tag{97}$$

with the sum indicating that the periodized noise is correlated with copies of itself from the future and the past. Thus in (96), all terms vanish except for when $mk - p = (nk - q) + \ell\widehat{N}$ or equivalently $q = p - (m-n)k + \ell\widehat{N}$. We then have

$$\widehat{R}_{m,n} = A^2 \frac{\Delta^2}{k^2} \sum_{p=0}^{\widehat{N}-1} \widehat{g}_p^{\{k\}} \left[\widehat{g}_{p-(m-n)r}^{\{k\}} + \widehat{g}_{p-(m-n)r+\widehat{N}}^{\{k\}}\right]^* \tag{98}$$

for the terms in the $N \times N$ covariance matrix $\widehat{\mathbf{R}}$. The second term arises from the Green's function interacting with a copy of itself shifted by $\widehat{N}$ due to the periodization of the noise, and is expected to be much smaller than the first term. Note that contributions from negative $\ell$ do not appear due to the fact that $\widehat{g}_n^{\{k\}}$ vanishes for negative $n$; but all contributions from $\ell > 1$ also vanish because $\widehat{g}_n^{\{k\}}$ vanishes for $n > \widehat{N} - 1$.

The advantage to this approach is that (95) is a discrete, periodic convolution that can be implemented using a Fast Fourier Transform in $O(\widehat{N} \log \widehat{N})$ operations; this is approximately $O(N \log N)$ since $\widehat{N} \approx N$. Making this $O(N)$, through a time-domain convolution with a Green's function that is truncated after a short time, is not feasible unless the Green's function decays very quickly, on account of blurring artifacts appearing in the spectrum due to the truncation. Our approach uses a Green's function that is the entire length of the desired sample in order to minimize such truncation effects.

If desired, the matrix $\widehat{R}_{m,n}$ in (98) can be computed in order to explicitly check the errors in computing the covariance matrix, although this will necessarily slow down the algorithm. From (77), the terms in the true, discretely sampled autocovariance


matrix are given exactly by

$$R_{m,n} = A^2 \int_0^{\hat{T}} g(s)g^*(s-(m-n)\Delta)\,ds + A^2 \int_{\hat{T}}^{\infty} g(s)g^*(s-(m-n)\Delta)\,ds \tag{99}$$

where $\hat{T} = (\hat{N}-1)\Delta$. We may observe that discretizing the first integral corresponds to the first summation in (98). There are therefore three error terms between $R_{m,n}$ and $\widehat{R}_{m,n}$: errors associated with this discretization, which are minimized by choosing the oversampling rate $k$; and errors from the second integral in (99) and the second summation in (98), both of which are minimized by choosing $N_\epsilon$ sufficiently large.

As an example, in Fig. 8 we present spectra of 25 samples of Matérn processes generated using both the Cholesky decomposition and the fast Green's function algorithm. The spectrum for each realization is computed using Thomson's adaptive multitaper algorithm (Thomson, 1982; Park et al., 1987) using 15 orthogonal Slepian tapers having a time-bandwidth product of eight. This approach employs frequency-domain smoothing to the extent that it can be achieved without the expense of broadband bias.

No substantial difference between spectra computed with the two different algorithms is seen over many decades of structure, indicating that fast algorithm is able to simulate the Matérn process to a very high degree of accuracy. In generating this plot, the Green's function algorithm executes 16–19 times faster than the Cholesky algorithm on a Mac laptop. Note that this method does not depend on any special properties of the Matérn process, apart from the particular definition of the cutoff time $T_\epsilon$ for the initial time period (93). The method is therefore suitable for any Gaussian random process having a decaying and sufficiently smooth autocovariance for which the Green's function has an analytic expression.

## 6 Application

This section presents the details of an application of the Matérn process to modeling particle velocities in a numerical simulation of two-dimensional fluid turbulence, a preview of which was presented in Section 2.5.

### 6.1 Numerical simulation of 2D turbulence

A system called *forced-dissipative quasigeostrophic turbulence* is created by integrating an equation for the streamfunction $\Phi(x,y,t)$. For nondivergent flows, the streamfunction is a scalar-valued quantity at each point giving the velocity components through $U(x,y,t) = -\frac{\partial}{\partial y}\Phi$ and $V(x,y,t) = \frac{\partial}{\partial x}\Phi$. The equation to be integrated is

$$\frac{\partial}{\partial t}\left(\nabla^2\Phi - \Phi/L_D^2\right) + J(\Phi, \nabla^2\Phi) = F - D \tag{100}$$

where $J(a,b) \equiv \frac{\partial a}{\partial x}\frac{\partial b}{\partial y} - \frac{\partial b}{\partial x}\frac{\partial a}{\partial y}$ is the Jacobian operator, $L_D$ is a spatial scale termed the deformation radius, $F$ is a forcing function, and $D$ is a damping. This equation is derived from a conservation law following particle trajectories. This simple system is considered an idealized representation of turbulence in planetary fluid dynamics, on scales large enough that the rotation of the planet is important, but not so large that the planet's curvature needs to be taken into account.



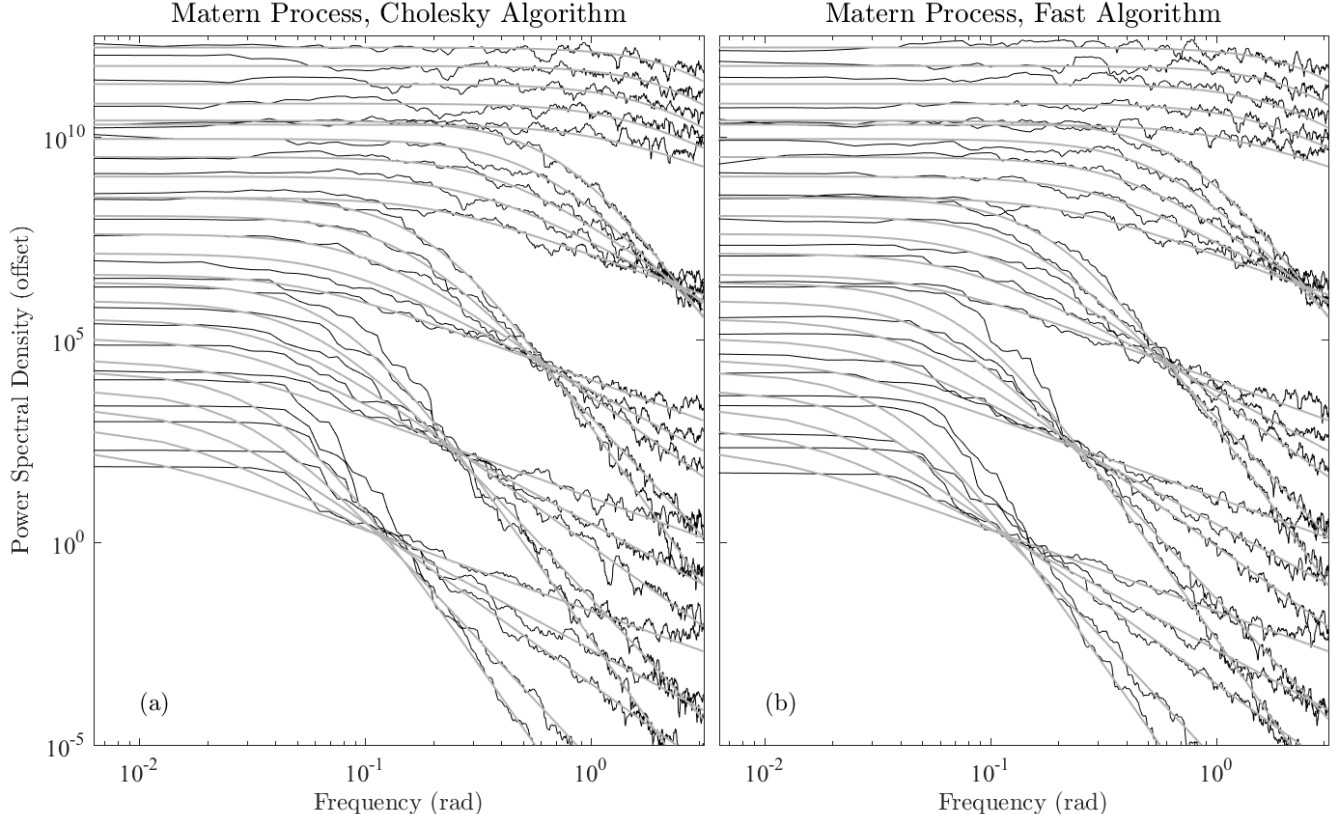

**Figure 8.** A comparison of the spectra of simulated unit-variance Matérn process having twenty-five different $(\alpha, \lambda)$ values for (a) the Cholesky decomposition algorithm and (b) the fast generation algorithm presented in Section 5.3. The process samples are each 1000 points long, with the sample interval $\Delta$ set to unity; frequency is thus given in radians. Black curves show the multitaper spectral estimates, as described in the text, while gray curves are the theoretical spectral forms. Successive spectral plots have been offset in the vertical by a factor of $\sqrt{10}$ for presentational clarity. The five lines within each group correspond to the five $\alpha$ values 1, 1.5, 2, 3, and 4. The five groups correspond to different values of $\lambda$, with $\lambda$ equal to 0.01, 0.02, 0.05 0.2, or 1 times the value of $\alpha$ for each curve, proceeding from bottom to top. Only positive frequencies are shown, as the theoretical spectra at negative frequencies are identical. Simulated spectra from the $O(N \log N)$ fast algorithm and those from the $O(N^3)$ Cholesky algorithm are found to be virtually identical.

An integration of (100) is carried at $1024^2$ resolution in a doubly periodic domain of dimension $2500 \times 2500$ km. As is typical in such problems, the forcing $F$ consists of random fluctuations of a particular spatial scale imposed everywhere in the domain at each time step. The characteristic forcing scale of 117 km is chosen here such that the scale of the forcing is intermediate between the grid scale and the domain scale. The damping is chosen to take the form $D = r\nabla^2\Phi$ where $r$ is set to $1.5 \times 10^{-8}$ s$^{-1}$. After an initial spin-up period, during which an equilibration of energy levels is achieved, the simulation is run for three years or 3*365=1095 days.



A snapshot of current speed from the first day of the simulation after the end of the spin-up period is shown in the left panel of Fig. 1. As mentioned previously, the circular areas of high speed represent long-lived vortices (see e.g McWilliams, 1990b; Scott and Dritschel, 2013), which are not the subject of this study. Instead we are interested in the behavior of particles that inhabit the spaces between the vortices.

5   The analysis here is based on a set of 1024 particle trajectories that are tracked throughout this experiment, shown in the right panel of Fig. 1. The trajectories are output at high temporal resolution, then decimated to 6 hourly sampling and first central differenced to produce velocity. Position and velocity records are then decimated again to daily resolution, which we find to be sufficient to capture meaningful variability. One-half of the trajectories are then discarded in order to exclude those most directly effected by vortices, as described next, leaving 512 trajectories of length 1095 to be analyzed.

10   The simplest way to remove the effects of vortices is simply to discard those trajectories which conspicuously exhibit the effects of vortex trapping. A common measure of the impact of vortices on a given trajectory is the so-called spin parameter (Sawford, 1999; Veneziani et al., 2005b, a), defined as

$$\overline{\Omega} \equiv \frac{\overline{u(t)\frac{\mathrm{d}}{\mathrm{d}t}v(t) - v(t)\frac{\mathrm{d}}{\mathrm{d}t}u(t)}}{\overline{u^2(t) + v^2(t)}} = \frac{\overline{\Im\left\{z^*(t)\frac{\mathrm{d}}{\mathrm{d}t}z(t)\right\}}}{\overline{|z(t)|^2}} \tag{101}$$

in which "$\Im$" is the imaginary part. In our implementation, these time derivatives are adequately approximated by first central differences at daily resolution. The overbar here is a temporal average over the extent of a trajectory; note that since the mean velocity is zero, the denominator is the velocity variance along the trajectory.

We take the modulus of the time-averaged spin, $|\overline{\Omega}|$, as a measure of the overall impact of vortices. Because of the persistence of particles within vortices, it is unlikely that a small value of $|\overline{\Omega}|$ would result from cancellation of positive and negative contributions within the same time series for the three-year lengths we consider. Conservatively, we keep the half of the 1024 trajectories having the lower values of spin magnitude. The resulting 512 trajectories, offset to begin at the origin in Fig. 2a, exhibit a meandering character in addition to their dispersion. The omitted trajectories typically present a dense and regular looping structure, some of which may be seen in the right-hand panel of Fig. 1.

## 6.2   Frequency-domain maximum likelihood

This section describes the method by which the Matérn parameters are estimated from a finite data sample, which necessitates some new notation. In reality one only observes a random process $z(t)$ at a finite set of discrete times $z[n] = z(n\Delta)$ separated by the time interval $\Delta$, and with $n = 0, 2, \ldots, N-1$. In this subsection, we use square brackets for time series which take discrete arguments, thereby distinguishing a discretely sampled time series $z[n]$ from its continuous analogue $z(t)$. Based on this sample $z[n]$, one wishes to estimate the parameters of stochastic model, conventionally denoted by the vector $\boldsymbol{\theta}$, which in the case of the Matérn model is $\boldsymbol{\theta} = (\sigma, \alpha, \lambda)$.

30   A standard approach to estimating the parameters is to do so in the frequency domain using a method called the Whittle likelihood (Whittle, 1953), as follows. The discrete Fourier transform of the length $N$ sequence $z[n]$ is given by

$$Z[m] \equiv \sum_{n=0}^{N-1} z[n] e^{-\mathrm{i}2\pi mn/N} \tag{102}$$





for $m = 0, 1, 2, \ldots, (N-1)$. The squared modulus of this sequence of $N$ Fourier coefficients defines a spectral estimate known as the *periodogram*

$$\widehat{S}_{zz}[m] \equiv \frac{1}{N} |Z[m]|^2. \tag{103}$$

This is to be compared with the discretely sampled theoretical spectrum (59) for a particular value of the parameters $\boldsymbol{\theta}$

$$S_{zz}^{\boldsymbol{\theta}}[m] = S_{zz}^M \left( \frac{2\pi m}{N\Delta} \right) = \frac{\lambda^{2\alpha-1}}{c_\alpha} \frac{\sigma^2}{\left[ \left( \frac{2\pi m}{N\Delta} \right)^2 + \lambda^2 \right]^\alpha} \tag{104}$$

where $2\pi m/(N\Delta)$ is recognized as the $m$th Fourier frequency.

The model parameters are estimated by finding the value of $\boldsymbol{\theta}$ that maximizes the so-called Whittle log-likelihood

$$\ell(\boldsymbol{\theta}) = -\sum_{m \in \mathcal{F}} \left\{ \ln S_{zz}^{\boldsymbol{\theta}}[m] + \frac{\widehat{S}_{zz}[m]}{S_{zz}^{\boldsymbol{\theta}}[m]} \right\} \tag{105}$$

in which $\mathcal{F}$ is a set of integers indicating the Fourier frequencies over which the fit is to be applied. For example, $\mathcal{F}$ could be chosen to be $m = 0, 1, 2, \ldots, (N-1)$, in which case the fit will be applied to all frequencies.

In turns out to be the case that in the inference of parameters for a steep spectrum, such as we are dealing with here, this approach is inadequate as it ignores potentially significant effects associated with the finite sample size. In particular, *spectral blurring* associated with the periodogram can lead to quite incorrect slopes at high frequencies. Instead we use the *de-biased* Whittle likelihood method recently developed by Sykulski et al. (2015). In that approach, the periodogram $\widehat{S}_{zz}[m]$ in (105) is replaced with a *tapered* spectral estimate, and the theoretical spectrum $S_{zz}^{\boldsymbol{\theta}}[m]$ is replaced with the *expected* tapered estimate for a Matérn process characterized by the particular value of $\boldsymbol{\theta}$. The de-biased Whittle likelihood allows the parameters $\boldsymbol{\theta}$ to be more accurately estimated, as it correctly accounts for the effect of spectral leakage as well as aliasing.

### 6.3 Model realizations

Here we give details on how the realizations shown in Fig. 2b–d have been created. First, in preparing Fig. 3, tapered spectral estimates as well as periodogram estimates are formed. As discussed in Section 2.5, for data tapers we use the lowest-order Slepian taper (Slepian, 1978; Thomson, 1982; Park et al., 1987; Percival and Walden, 1993) with the time-bandwidth product set to 10. The average over all time series, and over both sides of the frequency spectrum, are shown for both estimates. In comparison with the tapered estimates the periodogram is seen to accurately estimate the spectrum, over only about half of the dynamic range. This illustrates the potentially severe problems with using the standard Whittle likelihood for parameter inference involving steep spectra, and motivates our use of the de-biased method.

After forming the tapered spectral estimate for each of the 512 turbulence velocity time series, we apply the de-biased Whittle likelihood to infer the best fit Matérn parameters for each time series. Here the frequency set $\mathcal{F}$ is chosen to include frequencies up to 1.5 radians per day, as this corresponds to the upper limit of apparent structure in the spectra. For each set of parameters, we generate a realization of a Matérn process having these properties as described in Section 5, and then cumulatively sum these velocity time series to produce the trajectories shown in Fig. 2b. Estimation of the spectra for these





Matérn realizations in the same manner as for the turbulence data leads to the black dashed line shown in Fig. 3, which is seen to be a very close match to the velocity spectra for the particle trajectories from the turbulence simulation.

To generate the trajectories shown in Fig. 2c and Fig. 2d, we proceed as values. The parameters value from the fit to the Matérn form are converted to a diffusivity through $\kappa = \frac{1}{4}\sigma^2/(\lambda c_\alpha)$, which is then used to scale realizations of white noise. The spectra of the associated velocities in Fig. 3c are seen as matching the low-frequency values of the Lagrangian velocity spectra from our turbulence simulation. Cumulatively summing these velocities produces the trajectories in Fig. 2c; note that these trajectories therefore consist of discrete samples of standard Brownian motion. These are seen to match well the dispersion characteristics of the turbulence trajectories, but to have far too high a degree of small-scale roughness.

For the power-law realizations, we cannot employ fractional Brownian motion because the observed slopes—which in this simulation is steeper than those found in the ocean—are outside the fBm range. Instead we use the implied spectral amplitudes $A^2 = \sigma^2 \lambda^{2\alpha-1}/c_\alpha$ and slope parameters $\alpha$ from the Matérn fit to fix the properties of a different Matérn process having a very small damping value, chosen as $\lambda = 2\pi/T$ where $T$ is the record duration. Realizations are then generated and cumulatively summed to give the trajectories shown in Fig. 2d. As mentioned before, these have vastly too much energy on account of extending the high-frequency slope across all frequencies. The flattening of the estimated spectrum for these realizations seen in seen in Fig. 3 is a result of the extreme dynamic range hitting the limit of numerical precision.

The point of the application is to show that Matérn process provides an excellent match to the turbulence turbulence data. This opens the door to investigating a number of interesting questions regarding the distributions and interpretations of those parameters, which must, however, be left to the future.

## 7 Discussion

This paper has examined the Matérn process, a little-used random process that we have shown to be equivalent to damped fractional Brownian motion (fBm). The damping is shown to be essential for permitting the phenomenon of diffusivity within the temporal integral of the process, referred to here as the *trajectory*, which disperses from its initial location at a constant rate. The rate of diffusion of the trajectory is given by the value of the spectrum of the process at zero frequency. At higher frequencies, the spectrum transitions to a power-law slope, like fBm, with the location of this transition being controlled by the damping parameter.

Because damping is a common feature in physical systems, the Matérn process may be valuable in describing time series which, when observed over shorter time intervals, appear to consists of fractional Brownian motion. The addition of a spin parameter leads to a still more general process that satisfies the stochastic integral equation for a damped fractional oscillator forced by continuous-time white noise, and that encompasses the standard Matérn process as well as the complex (Jeffreys, 1942; Arató et al., 1999) and standard (Uhlenbeck and Ornstein, 1930) Ornstein-Uhlenbeck processes within a single larger family. A simple algorithm for generating approximate realizations of this 'oscillatory Matérn' process in $O(N \log N)$ operations was presented.





A categorization of stochastic processes as *diffusive*, *subdiffusive*, and *superdiffusive* was proposed, depending upon their value at zero frequency. These categorizations refer to the nature of the dispersion experienced by the trajectory associated with the process, assuming that the integral of the process is well defined. This categorization is related to, yet distinct from, the conventional designation of a random process as short-memory or long-memory (Beran, 1994). We have argued that the

diffusivity categorization may prove to be a powerful way to describe stochastic processes in general. The Matérn process considered here is an example of a stationary diffusive process, and modified versions of it were shown to provide examples for five of the six possible combinations of memory and diffusiveness. It was demonstrated that the exception, a short memory superdiffusive process, cannot exist.

The Matérn process was found to provide an excellent match to velocity time series from those particle trajectories in

forced/dissipative two-dimensional fluid turbulence that were not directly influenced by the presence of vortices. This is an important contribution, since we show that a power-law process such as fBm cannot hope to capture the diffusive behavior. Despite its simple three-parameter form, trajectories associated with the Matérn process were seen to be visually virtually indistinguishable from those from the numerical model. This suggests that the Matérn form may prove useful for describing similar trajectories taken by instruments tracking the ocean currents. Such 'Lagrangian data' is one of the main windows into

observing the ocean circulation, yet surprisingly little work has been done to analyze the velocity spectra in major Lagrangian datasets (Rupolo et al., 1996; Elipot and Lumpkin, 2008). Apart from Rupolo et al. (1996), the spectral slope in oceanographic Lagrangian data is almost completely unexplored, although it is implicit in several fractal dimension studies (Osborne et al., 1989; Sanderson et al., 1990; Sanderson and Booth, 1991; Summers, 2002).

In this paper, we have taken essentially an observational approach, and sought to fit a parametric model to the trajectories as

a descriptive analysis, without requiring a physical justification. A next step is to attempt to understand this model on physical grounds. A number of researchers have attempted to derive forms for the Lagrangian velocity spectrum (or, equivalently, the autocovariance function) under simplified dynamical assumptions (Griffa, 1996; Weiss et al., 1998; Majda and Kramer, 1999; Berloff and McWilliams, 2002; Veneziani et al., 2005a; Majda and Gershgorin, 2013). One promising avenue of comparison is with the work of Berloff and McWilliams (2002), who derive dynamical models roughly equivalent to integral orders of the

Matérn process. Another is with Majda and Kramer (1999), see their Section 3.1.2, who construct idealized velocity fields that give rise to the diffusive, subdiffusive, and superdiffusive regimes of Lagrangian behavior. Exploring the relationship of the Matérn form to these dynamical models is a promising direction for future research.

**Acknowledgements**

The work of J. M. Lilly and J. J. Early was supported by award #1031002 from the Physical Oceanography program of the

United States National Science Foundation. The work of A. M. Sykulksi was supported by a Marie Curie International Outgoing Fellowship. The work of S. C. Olhede was supported by awards #EP/I005250/1 and #EP/L025744/1 from the Engineering and Physical Sciences Research Council of the United Kingdom, and by award #682172 from the European Research Council.





## Appendix A:  A freely available software package

All software needed to carry out the analyses described in this paper, and to generate all figures, is distributed as a part of a freely available toolbox of Matlab functions.[3] This toolbox, called `jLab`, is available at http://www.jmlilly.net and is distributed under a Creative Commons license.

The particular package of routines most relevant to this paper is called `jMatern`, and includes the following functions: `materncov`, `maternspec`, and `maternimp`, which implement the Matérn autocovariance function, spectrum, and impulse response or Green's function, respectively; `maternoise`, which generates realizations of the Matérn process using either the standard Cholesky decomposition method, or the fast generation method described in Section 5; `maternfit`, which performs a parametric spectral fit for the Matérn process and a number of variations, using the de-biased Whittle likelihood method

discussed in Section 6.2; and `blurspec`, which accounts for the blurring and/or aliasing of the theoretical spectrum associated with truncation of a continuous random process or the tapering of a finite sample. All functions support the oscillatory Matérn process as well as the standard Matérn process. Finally, `makefigs_matern` generates all figures in this paper based on model output that can be downloaded from http://www.jmlilly.net.

## Appendix B:  The form of the fBm coefficient

The usual form of the coefficient for fractional Brownian motion, in terms of the Hurst parameter $H = \alpha - 1/2$, is

$$V_H \equiv \frac{\Gamma(1-2H)\cos(\pi H)}{\pi H} \tag{B1}$$

see Barton and Poor (1988). In terms of the slope parameter $\alpha$, this becomes

$$V_\alpha \equiv \frac{\Gamma(2-2\alpha)\sin(\pi\alpha)}{\pi(\alpha-1/2)} \tag{B2}$$

which can be expressed in a more symmetric form as follows. First we expand the denominator using $\Gamma(1+\nu) = \nu\Gamma(\nu)$ or

$\nu = \Gamma(1+\nu)/\Gamma(\nu)$ with $\nu = \alpha - 1/2$, giving

$$V_\alpha = \frac{\Gamma(2-2\alpha)\Gamma\left(\alpha-\frac{1}{2}\right)\sin(\pi\alpha)}{\pi\Gamma\left(\alpha+\frac{1}{2}\right)}. \tag{B3}$$

The so-called *reflection* and *duplication* theorems for the gamma function are, respectively,

$$\sin(\pi\nu) = \frac{\pi}{\Gamma(\nu)\Gamma(1-\nu)} \tag{B4}$$

$$\Gamma(2\nu) = \frac{1}{\sqrt{\pi}}2^{2\nu-1}\Gamma(\nu)\Gamma\left(\nu+\frac{1}{2}\right) \tag{B5}$$

see 6.1.17 and 6.1.18 on p. 256 of Abramowitz and Stegun (1972). Applying the later to both $\Gamma(2\alpha)$ and $\Gamma(2-2\alpha)$ gives their product as

$$\Gamma(2\alpha)\Gamma(2-2\alpha) = \frac{1}{\pi}\Gamma(\alpha)\Gamma\left(\alpha+\frac{1}{2}\right)\Gamma(1-\alpha)\Gamma\left(\frac{3}{2}-\alpha\right) \tag{B6}$$

---

[3]To be made available upon acceptance for publication.




in which all powers of two exactly cancel. Employing the reflection theorem with $\nu = \alpha$, this becomes

$$\Gamma(2\alpha)\Gamma(2-2\alpha) = \frac{\Gamma\left(\alpha + \frac{1}{2}\right)\Gamma\left(\frac{3}{2} - \alpha\right)}{\sin(\pi\alpha)} \tag{B7}$$

and substituting this into (B3) leads to (26), as claimed.

Now, using the reflection formula (B4) together with a trigonometric identity we find

$$-\cos(\pi\alpha) = \sin\left(\pi\alpha - \frac{\pi}{2}\right) = \frac{\pi}{\Gamma\left(\alpha - \frac{1}{2}\right)\Gamma\left(\frac{3}{2} - \alpha\right)} \tag{B8}$$

and therefore

$$-\frac{1}{\cos(\pi\alpha)\Gamma(2\alpha)} = \frac{1}{\pi}\frac{\Gamma\left(\alpha - \frac{1}{2}\right)\Gamma\left(\frac{3}{2} - \alpha\right)}{\Gamma(2\alpha)} = V_\alpha. \tag{B9}$$

This establishes that the coefficient found in the Abel limit (37) in Section 3.3 is the same as $-V_\alpha/2$.

**Appendix C: Fractional Gaussian noise**

Define the difference of a fractional Brownian motion process at one time and itself a different time as

$$z_\Delta(t) \equiv z(t + \Delta) - z(t) \tag{C1}$$

which will be explicitly labeled by the time interval $\Delta$ for clarity. The resulting process is called *fractional Gaussian noise* or fGn (Mandelbrot and Van Ness, 1968; Mandelbrot and Wallis, 1969; Percival and Walden, 1993). While it is more usual to sample the process defined by (C1) at regular intervals, here we will examine the properties of the continuous process.

The autocovariance function for continuous fractional Gaussian noise will be denoted as

$$R_{zz,\Delta}^{fGn}(t,\tau) \equiv \mathrm{E}\left\{z_\Delta(t + \tau)\, z_\Delta^*(t)\right\} \tag{C2}$$

and this expands to give

$$R_{zz,\Delta}^{fGn}(t,\tau) = R_{zz}^{fBm}(t,\tau) + R_{zz}^{fBm}(t+\Delta,\tau) - R_{zz}^{fBm}(t,\tau+\Delta) - R_{zz}^{fBm}(t+\Delta,\tau-\Delta). \tag{C3}$$

Substituting the form of the fBm autocovariance (21), cancellations occur, leading to

$$R_{zz,\Delta}^{fGn}(\tau) \equiv R_{zz,\Delta}^{fGn}(t,\tau) = \frac{V_\alpha}{2}A^2\left[|\tau + \Delta|^{2\alpha-1} + |\tau - \Delta|^{2\alpha-1} - 2|\tau|^{2\alpha-1}\right] \tag{C4}$$

where our notation is modified to reflect the fact that the autocovariance is independent of $t$. Fractional Gaussian noise is therefore a stationary process. On account of the self-similar scaling of the fBm autocovariance function (41), one finds

$$R_{zz,\Delta}^{fGn}(\tau) = \Delta^{2\alpha-1}R_{zz,1}^{fGn}(\tau/\Delta) \tag{C5}$$

so that we may without loss of generality set $\Delta = 1$. For convenience we let $\tilde{\tau} \equiv \tau/\Delta$ be a nondimensional time offset.





The expression (C4) may be compared with (5.2) of Mandelbrot and Van Ness (1968), who permitted the durations of the two increments to differ. Our expression differs from that in Mandelbrot and Van Ness (1968) because we have chosen to apply the similarity scaling to remove the increment *duration* rather than the *separation*, for reasons to become apparently shortly; "$T$" in Mandelbrot and Van Ness (1968) refers to what we call $\tau$ here.

The normalized fGn covariance function $R_{zz,1}^{fGn}(\tilde{\tau})/V_\alpha$ with the amplitude $A$ set to unity is shown in Fig. 9. Because fGn will generally be sampled, we are typically interested only in time offsets $\tau$ that exceed the sample interval $\Delta$, corresponding to $\tilde{\tau} > 1$. Analyzing $R_{zz,1}^{fGn}(\tilde{\tau})$ using (C4), one sees that for $\tilde{\tau} > 1$ it obtains a maximum value of unity at $\alpha = 3/2$, while it vanishes both for $\alpha = 1/2$ and $\alpha = 1$. It is found that $R_{zz,1}^{fGn}(\tilde{\tau})$ is positive for $\alpha > 1$, and negative for $\alpha < 1$, see Mandelbrot and Van Ness (1968). The maximum positive value is at $\alpha = 3/2$ for all $\tilde{\tau}$, but the maximum negative value occurs at some

intermediate value of $\alpha$ in the range $(1/2, 1)$. For any fixed $\alpha$, increasing $\tilde{\tau}$ leads to absolute values of $R_{zz,1}^{fGn}(\tilde{\tau})$ that decay toward zero.

The behavior of the fractional Gaussian noise covariance function allows us to discuss the property of *persistence*. For $\alpha > 1$, fGn exhibits positive correlations, such that positive values will tend to be followed by positives value and negative values by negative values. However, for $\alpha < 1$, fGn is *anti-persistent*, and positive values will tend to be followed by negative values and

vice-versa. Note that $R_{zz,1}^{fGn}(\tilde{\tau})$ is not symmetric about $\alpha = 1$: the most positive correlations occur at $\alpha = 3/2$, but the most negative correlations do not occur at $\alpha = 1/2$. This may perhaps be seen as reflecting a difference between persistence and anti-persistence. Values of the same sign can follow one another indefinitely, for any time scale; but the same cannot be true for values of the opposite sign.

The persistence transition in fractional Gaussian noise at $\alpha = 1$ is reflected in the behavior of fractional Brownian motion

seen in Fig. 5. Values of $\alpha > 1$ coincide with the tendency for the process to systematically drift away from an initial value, as differenced versions of the process will tend to keep contributing perturbations of one particular sign. Similarly, for $\alpha < 1$, the anti-correlations of the differenced process tend to act to restore fBm toward a baseline, and therefore these process are more closely distributed around the mean value of zero. The important point is that for fractional Brownian motion, the spectral slope can be seen as being linked to the degree of persistence or anti-persistence associated with a differenced version of the process.

The memory of fractional Gaussian noise may be determined as follows. The fGn autocovariance (C4) can be rewritten as

$$R_{zz,\Delta}^{fGn}(\tau) = \frac{V_\alpha}{2} A^2 \tau^{2\alpha-1} \left[ |1 + \Delta/\tau|^{2\alpha-1} + |1 - \Delta/\tau|^{2\alpha-1} - 2 \right] \tag{C6}$$

after pulling out the factor of $\tau^{2\alpha-1}$. Employing the binomial expansion, $(1+x)^\gamma = 1 + \gamma x + \frac{1}{2}\gamma(\gamma-1)x^2 + O(x^3)$ for small $x$, cancellations occur and we find

$$R_{zz,\Delta}^{fGn}(\tau) = \frac{V_\alpha}{2} A^2 \Delta^2 (2\alpha-1)(2\alpha-2)\tau^{2\alpha-3} + O\left(\tau^{2\alpha-2}\right) \tag{C7}$$

for the asymptotic behavior at large $\tau$. Recall from Section 2.4 that a *long-memory* stationary process is one for which the long-time behavior of the autocovariance function behaves as $R_{zz}(\tau) \sim |\tau|^{-\mu}$ for $0 < \mu \leq 1$. For fGn we have $\mu = 3 - 2\alpha$ with $1 < \alpha < 3/2$, thus fractional Gaussian noise is a long-memory process.



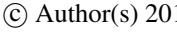

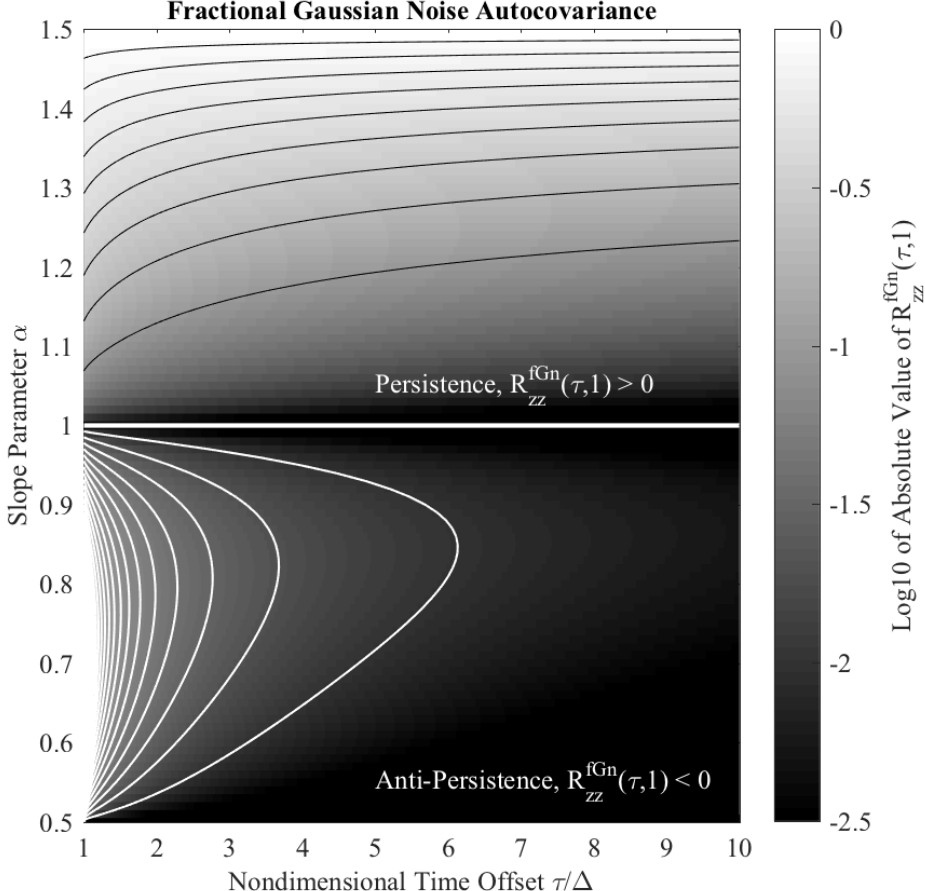

**Figure 9.** The fractional Gaussian noise autocovariance function $R_{zz,1}^{fGn}(\widetilde{\tau})$, as defined in (C4), here normalized by dividing by $V_\alpha$ and with $A = 1$. The time axis is interpreted as the normalized time $\tilde{\tau} = \tau/\Delta$. The shading shows $\log_{10}$ of the magnitude of the normalized autocovariance function, which obtains a maximum of unity at $\alpha = 3/2$ for all $\widetilde{\tau}$. A sign change occurs at $\alpha = 1$, with positive values at higher $\alpha$ and negative values at lower $\alpha$. Black lines are contours of positive values, with a contour interval of 0.1 beginning at zero, while thin white lines are contours of negative values with an interval of 0.01. The heavy white curve is the zero contour at $\alpha = 1$.

## Appendix D: The Matérn autocovariance for small $\tau$

In this appendix we derive the form of the small-$\tau$ behavior of the Matérn autocovariance function, as was apparently first done by Goff and Jordan (1988). Here we follow those authors, paying particularly close attention to the $\alpha$ range over which the result is valid. For this we will make use of the identity 9.6.2 of Abramowitz and Stegun (1972)

$$5 \quad \mathcal{K}_\nu(\tau) = \frac{1}{2}\pi \frac{\mathcal{I}_{-\nu}(\tau) - \mathcal{I}_\nu(\tau)}{\sin(\nu\pi)} \tag{D1}$$



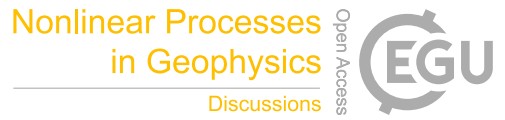

together with the series expansion 9.6.10 of Abramowitz and Stegun (1972)

$$\mathcal{I}_\nu(\tau) = \left(\frac{1}{2}|\tau|\right)^\nu \sum_{n=0}^{\infty} \frac{\left(\frac{1}{2}\tau\right)^{2n}}{n!\,\Gamma(n+1+\nu)}. \tag{D2}$$

Employing the reflection formula (B4), these combine to give

$$\tau^\nu \mathcal{K}_\nu(\tau) = \frac{1}{2}\Gamma(1-\nu)\Gamma(\nu)\left[2^\nu \sum_{n=0}^{\infty} \frac{\left(\frac{1}{2}\tau\right)^{2n}}{n!\,\Gamma(n+1-\nu)} - \frac{|\tau|^{2\nu}}{2^\nu}\sum_{n=0}^{\infty} \frac{\left(\frac{1}{2}\tau\right)^{2n}}{n!\,\Gamma(n+1+\nu)}\right] \tag{D3}$$

5 and gathering the terms for $n=0$, one finds

$$\tau^\nu \mathcal{K}_\nu(\tau) = \frac{1}{2}\Gamma(\nu)2^\nu\left[1 - \left(\frac{|\tau|}{2}\right)^{2\nu}\frac{\Gamma(1-\nu)}{\Gamma(1+\nu)}\right] + \sum_{n=1}^{\infty} \tau^{2n}\left[c_n + d_n|\tau|^{2\nu}\right] \tag{D4}$$

where $c_n$ and $d_n$ are constants describing the behavior proportional to $\tau^{2n}$ and $|\tau|^{2n+2\nu}$, respectively.

For small $|\tau|$, the term on the first line of (D4), which is proportional to $|\tau|^{2\nu}$, dominates the first term in the summation on the second line, proportional to $\tau^2$, provided that $\nu < 1$; all other terms are then smaller still. Since $\nu$ in these expressions is

10 related to $\alpha$ in the Matérn autocovariance function through $\nu = \alpha - 1/2$, this domination occurs for $\alpha < 3/2$, and we obtain the asymptotic behavior (62) for $|\tau| \ll 1/\lambda$. Note that this result is only valid for $\alpha < 3/2$, a point that does appear to have been mentioned by Goff and Jordan (1988). For larger values of $\alpha$, the smallest power of $\tau$ in (D4) is the $\tau^2$ term on the second line of (D4), which therefore dominates.



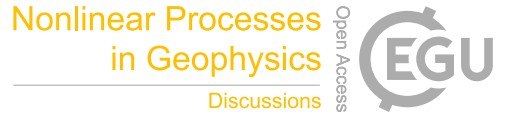

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
