# Peer review of "Fractional Brownian motion, the Matérn process, and stochastic modeling of turbulent dispersion"

_Nonlinear Processes in Geophysics, 2017_

## Referee Comment (RC1) · P. D. Ditlevsen (Referee) · 19 Apr 2017

This is a well written review of the Fractional Brownian motion (fBm) and the less well known Matérn process. The model is applied to the case of particle dispersion in 2D turbulent flow. The review is clear and with the relevant degree of detail to be readable by non-experts in stochastic processes.

Concepts of short and long memory and of diffusive, sub-diffusive and super-diffusive processes are well described as is the whole concept of diffusivity for stochastic processes, which is less known in the "time series community". The treatment of diffusivity, auto-covariance and corresponding spectra is illuminating.

[Figure]

The relevance of the Matérn process is that it provides a way of constructing a stationary process, which on short time scales is similar to the fBm and on long time scales is similar to a white noise process.

The paper is very long, so I propose that the authors perhaps consider rewriting into two back-to-back papers, with the second being the application to turbulent dispersion. Furthermore, for better overview, some of the results, especially in section 4, could be moved to Appendices. I leave that to the authors to decide and recommend publication with pleasure.

Minor points/typos:

P10, L25-30: seems to be irrelevant paragraph, including 4 self-references.

P14, Figure 3: It would be useful to indicate the slope of the "Power Law line", and indicate the relation to the turbulence scaling theory (if any).

P19, L3: faveraging -> averaging

P20, L27: It would be useful with a definition of a Gaussian process in this context, and a little more explanation of why the "original process is Gaussina as well as zero mean".

P26, L6: To me it seems more logical to say "globally white" rather than "locally white", in the sense that points separated in time by $T \gg \lambda^{-1}$ are independent.

P27, L3: This is unclearly written: It is $d^2/d\omega^2(lnS) = 0$ for $|\omega| = \lambda$.

P33, L16: $\Delta \to \Delta t$.

---

## Referee Comment (RC2) · Anonymous Referee #2 · 11 May 2017

The paper contains a collection of facts about the fractional Brownian motion and the Matérn process. These processes are used to model the velocity of particles moving in the complex plane. The paper explores the properties of the spectral density and their relation to the process diffusivity and memory. In Section 5 the authors propose a new approximate simulation algorithm for the Matérn process.

The main virtue of this paper is that it links statistical concepts and objects with physical ones, providing the intuition behind mathematical formulae and explaining their physical meaning. Mots of explanations are intuitively clear, calculations are detailed. This results in the size of the paper; the paper is very long and I recommend to split it into two papers. Sometimes the authors use objects before introducing them (for example,

the Matérn process or autocovariance function; please, see also the comments below).

**General remarks and questions**

- Statistically fBM and the Matérn process are very different in nature: fBM is not stationary and starts from 0; on the contrary, the Matérn process is stationary and its value at 0 value is random. Please clarify if this difference important for the physical model used in the paper.
- Page 3, line 1: please explain what 'damped version' of a process mean. This term is used many times throughout the paper, but it is never defined.
- Page 3, line 14: what parameters are meant here? Hurst index for fbm and a scale parameter for fbm? What is the third parameter of the Matérn process?
- Page 5, lines 15-16: in what sense the derivative (stochastically, pathwise, in *Lp* sense) is taken?
- Page 5, line 23: you use here the terms 'autocovariance function' and 'spectral density', but you define them only in the next subsection. The rearrangement of this subsection might be helpful for the reader.
- Page 7, line 5: please define the isotropy of a process to avoid the confusion with isotropy in spatial statistics, where it means the invariance of the process under rotation of coordinates.
- Page 9, line 20: please give the definition of the Matérn process and formula for its spectrum here. Is it a Gaussian process?
- Page 9, line 21: in what sense the derivative is taken?
- Page 10, line 17: what is periodic domain?

**NPGD**
- Page 13, line 11: please write the range of the parameters.
- Page 15, line 23: the equality sign here without the reference to Section 3.3 is misleading. In Section 2.3 you define the spectrum for stationary process. Fractional Brownian motion is not stationary, so at that stage it is not clear what this equality means.
- Pages 18-19, beginning of Section 3.3. it is worth mentioning that the spectrum of stationary process and its time-independent spectrum coincide. Thus, timeindependent spectrum is a more general notion than spectrum. Otherwise it is not clear, why one can compare the spectrum of stationary process and timeindependent spectrum of non-stationary process.
- Page 24, lines 13-14: the sentence is not clear. There are two separate integrals in equation (44). Why the integral in (47) cannot be split into two integrals?
- Page 25, lines 26-27: the following statement is not clear. The first is that there is no upper bound on  $\alpha$ , so processes can become still smoother than the  $\alpha = 3/2$  case that defines the upper limit of the slope parameter for fBm'. If  $\alpha_M > \alpha_{fBM}$ , where  $alpha_M$  is the smoothness parameter of the Matérn process and  $\alpha_{fBM}$  is the smoothness parameter of the fBM, it does not mean that the corresponding Matérn process is smoother than the corresponding fBM.
- It would be interesting to have the comparison of your simulation algorithm with known approximate algorithms (for example, spectral method). Note, that exact realizations for some values of  $\alpha$  can be generated fast using a modification of circulant embedding called cut-off circulant embedding, see ().
- Please explain how to choose parameters k and  $\hat{N}$  optimally. How can one estimate the error of the simulation for given k and  $\hat{N}$  ?
- Please explain why you do not use classical maximum likelihood estimation in the time domain. What is the benefit of using frequency-domain maximum likelihood? It would be interesting to see the box-plots and bootstrap confidence intervals for the estimated parameters of the times series generated by the proposed algorithm.
- For the reader's convenience, when citing books, please include the number of the relevant chapter or pages.

**Minor issues and typos**

- Page 3, line 2: 'a uniform rotation rate to is shown'
- Page 3, line 30: 'This paper was inspired by the need to a develop'
- Page 9, Table 1: "The term in the box is the Matérn process", you probably mean that the term in the box is the spectrum of the Matérn process. Please write the ranges for  $\lambda, \Omega, \alpha$ .
- Page 9, line 17: if a function is absolutely integrable, then it is integrable; not vice versa.
- Page 17, lines 5-6: the sentence "The exponent  $2\alpha$ -1 varies from ..." is very difficult to understand, however it has a very simple meaning, namely  $0 < 2\alpha$ -1 < 2.
- Page 19, line 3: 'faveraging'
- Page 27, lines 1-3: the sentence is unclear. What vanishes at  $|w| = \lambda$ ?
- Page 41, line 2: we proceed as values
- Page 41, line 16: turbulence turbulence.
**References**

Gneiting, T. and Ševčíková, H. and Percival, D. B. and Schlather, M. and Jiang, Y. *Fast and exact simulation of large Gaussian lattice systems in* ℝ2: *exploring the limits.* Journal of Computational and Graphical Statistics, 15, 483–501, 2006.

---

## Author Comment (AC1) · 5 Jul 2017

We would like to thank the reviewer, Peter Ditlevsen, for his careful review of our paper. We believe we have addressed all of his comments.

Major points

1. *This is a well written review of the Fractional Brownian motion (fBm) and the less well known Matérn process. The model is applied to the case of particle dispersion in 2D turbulent flow. The review is clear and with the relevant degree*

[Figure]

*of detail to be readable by non-experts in stochastic processes.*

Thank you for these positive comments. We are especially glad you found it suitable for non-experts in stochastic processes, as this was one of our main objectives.

2. *Concepts of short and long memory and of diffusive, sub-diffusive and super-diffusive processes are well described as is the whole concept of diffusivity for stochastic processes, which is less known in the "time series community". The treatment of diffusivity, auto-covariance and corresponding spectra is illuminating.*

The introduction of the concepts of diffusive, sub-diffusive, and super-diffusive as pertaining to types of *processes* is one of the main contributions of this paper. We are very glad to hear your found this illuminating.

3. *The paper is very long, so I propose that the authors perhaps consider rewriting into two back-to-back papers, with the second being the application to turbulent dispersion... I leave that to the authors to decide and recommend publication with pleasure.*

We believe it should remain as one long contribution. The main point of the paper is to connect the physical intuition with statistical concepts, and therefore we believe an integrated approach is preferable. Thank you for leaving this to our judgment.

4. *Furthermore, for better overview, some of the results, especially in section 4, could be moved to Appendices.*

We have done so. There are now four new Appendices: Appendix B, Diffusivity in terms of the spectrum, takes material from the previous section 2.3; Appendix C, Diffusiveness and memory, takes material from the previous section 2.4; Appendix D, The Rihaczek distribution of fractional Brownian motion, takes material from Section 3.3; and Appendix H, The Matérn impulse response function, takes material from Section 4.4.

Minor points

1. *P10, L25-30: seems to be irrelevant paragraph, including 4 self-references.* The point is actually important, but it has been greatly shortened and combined with the previous paragraph.

2. *P14, Figure 3: It would be useful to indicate the slope of the "Power Law line" and indicate the relation to the turbulence scaling theory (if any).*
   The power law line has been added. We are not aware of any turbulence theory for 2D frequency (as opposed to wavenumber) spectra; this has been noted in the text, as follows: "The spectral form is seen to provide an excellent match to the observed Lagrangian velocity spectra over roughly eight decades of structure. The high-frequency slope is seen to be roughly $|\omega|^{-8}$, a very steep slope. We are not aware of any physical theory to account for this, nor for the value of the damping parameter $\lambda$. Despite the fundamental role that the Eulerian *wavenumber* spectrum of velocity plays in turbulence theory, the Lagrangian *frequency* spectrum has received relatively little attention. Attempting to connect the observed form of this spectrum to physical principles is, however, outside the scope of the present paper."

3. *P19, L3: faveraging $\longrightarrow$ averaging.*
   Fixed.

4. *P20, L27: It would be useful with a definition of a Gaussian process in this context, and a little more explanation of why the "original process is Gaussian as well as zero mean".*
   Done.

5. *P26, L6: To me it seems more logical to say "globally white" rather than "locally white", in the sense that points separated in time by $T \gg \lambda$-1 are independent.*

The term "locally white" means that the spectrum is constant, or white, within some frequency range, but not everywhere. The meaning of this usage has been clarified.

6. *P27, L3: This is unclearly written: It is $d^2/d\omega^2(\ln S) = 0$ for $|\omega| = \lambda$. P33, L16: $\Delta \longrightarrow \Delta t$.*
   We were inconsistent in using $\Delta$ or $\Delta t$ for the sampling interval. Now, we have consistently used $\Delta$, without the $t$. Thank you for noticing this.

---

## Author Comment (AC2) · 5 Jul 2017

Responses to Reviewer 2

We would like to thank the reviewer for their very careful reading of the paper. This has helped us to clarify a number of subtle points, and helped us in our goal of balancing a high degree of mathematical rigor with intuitively clear explanations. We appreciate that reviewing a paper as long as this one is a substantial task, and we are very grateful for their effort and patience. We have answered all of the reviewer's concerns in detail and believe the paper has now been considerably improved.

Major points

1. *The main virtue of this paper is that it links statistical concepts and objects with physical ones, providing the intuition behind mathematical formulae and explaining their physical meaning. Mots of explanations are intuitively clear, calculations are detailed.*
   Thank you. This was our main objective, and we are very pleased to see that you believe we were successful.

2. *This results in the size of the paper; the paper is very long and I recommend to split it into two papers.*
   This is an issue with which we have wrestled many times in the course of preparing this paper. However, our conclusion is that in this case, a single long paper is appropriate for the topic, and is in fact an advantageous presentation. We will explain our reasoning here, together with the steps we have taken to address the reviewer's apparent concern that a paper of this length may place an undue burden on readers. Naturally if the reviewer, or editors, still feel strongly that the paper should be split in two, we would endeavor to do so, although it would take a fair bit of time and effort to accomplish.

   As the reviewer mentioned in the previous comment, the main contribution of this paper is the connection of statistical concepts with physical ones. Thus, we cannot split the physical part from the statistical part without losing the main point of the paper. Similarly, it would not be sensible to split the fBm and Matérn portions, because our goal is to connect them to each other. Thus, there is not an obvious way to split the paper in two. Either of these approaches would generate a huge number of cross-references and repeated material, making the

paper(s) less navigable, rather than more.

Instead, we have attempted to address the reviewer's concern by making the paper more readable and navigable through three modifications: moving more material to appendices, adding a table of contents, and introducing a signposting of what sections need to be read by whom.

Firstly, we have moved various portions of more technical material into four new appendices: Appendix B, Diffusivity in terms of the spectrum, takes material from the previous section 2.3; Appendix C, Diffusiveness and memory, takes material from the previous section 2.4; Appendix D, The Rihaczek distribution of fractional Brownian motion, takes material from Section 3.3; and Appendix H, The Matérn impulse response function, takes material from Section 4.4. This reduces the main text by about 3 pages in its current format, and considerably improves the readability of the paper.

Secondly, we have added a table of contents. If this is permissible within the journal format, we believe it could be highly useful to readers of this paper. The paper is intended to be a self-contained and comprehensive treatment of these stochastic processes, such that a researcher wishing to use the Matérn process does not need to first delve into the literature in order to understand fBm or fundamental concepts such as self-similarity. This self-containedness is a significant benefit offered by this paper, as in our view, there is nothing like it in the literature. Adding a table of contents would let the reader readily access those parts that are of interest, while skipping over any portions they are already familiar with, or which pertain to details they may feel they do not need to know. We feel this comprehensive organization would be more beneficial to readers then splitting the paper in two.

Finally, we have stated, at the beginning of each section, what the purpose of the section is and under what conditions a reader might be comfortable skipping that section.

3. *Sometimes the authors use objects before introducing them (for example, the Matérn process or autocovariance function; please, see also the comments below).*
We have rearranged things so that this does not occur.

General remarks and questions

1. *Statistically fBM and the Matérn process are very different in nature: fBM is not stationary and starts from 0; on the contrary, the Matérn process is stationary and its value at 0 value is random. Please clarify if this difference important for the physical model used in the paper.*
The following paragraph has been added at the end of Section 4: "The Matérn process and fBm differ in a qualitatively significant way: the former is stationary, while the latter is non-stationary. This difference can be seen as a consequence of the lack of damping in the latter case. In applications, we believe it would be unphysical to observe a process that remains nonstationary for all time scales. Rather, for sufficiently long observational periods, it is more likely that the process will eventually settle into stationary behavior. For the Matérn process, this occurs when the observational window is sufficiently long compared with the decay timescale $\lambda^{-1}$. Another difference is that the value of fBm at time $t = 0$ is fixed to zero, while that of the Matérn process is random. However, since it is common practice to remove the sample mean prior to analyzing a data time series, and/or to add a constant offset to a generated process, this distinction makes little practical difference for applications such as the one presented here."

2. *Page 3, line 1: please explain what 'damped version' of a process mean. This term is used many times throughout the paper, but it is never defined.*
We appreciate this comment, as we had not expected that this terminology might be unfamiliar to some readers. In the Introduction, we now say: "By 'damped version', we mean that the process is modified as would be expected if a physical damping were introduced into its stochastic differential or stochastic integral equation. This terminology, which draws upon intuition for damped and undamped oscillators from elementary physics, will be made more clear in Section 4.4." It is then discussed in detail in the last three paragraphs of Section 4.4, which are new, and in the first paragraph of Section 4.5.

3. *Page 3, line 14: what parameters are meant here? Hurst index for fbm and a scale parameter for fbm? What is the third parameter of the Matérn process?*
This has been clarified as: "More generally, the Matérn process adds a third parameter (damping) to the two parameters (amplitude together with spectal slope or the Hurst parameter) of fBm..."

4. *Page 5, lines 15-16: in what sense the derivative (stochastically, pathwise, in Lp sense) is taken?*
The need to deal with this subtlety is avoided by defining $r(t)$ as the integral of $z(t)$, rather than defining $z(t)$ as the derivative of $r(t)$.

5. *Page 5, line 23: you use here the terms 'autocovariance function' and 'spectral density', but you define them only in the next subsection. The rearrangement of this subsection might be helpful for the reader.*
Good point, this change has been made.

6. *Page 7, line 5: please define the isotropy of a process to avoid the confusion with isotropy in spatial statistics, where it means the invariance of the process under rotation of coordinates.*
The meaning is the same here. This section has been expanded for clarity as:

'[I]f one rotates the process counterclockwise through some some constant angle $\Theta$ by defining $\tilde{z}(t) \equiv e^{i\Theta}z(t)$, we have $R_{\tilde{z}\tilde{z}}(\tau) = R_{zz}(\tau)$, and the autocovariance function remains unchanged... With $\tilde{z}(t) \equiv e^{i\Theta}z(t)$ again being a rotated version the process, one finds $C_{\tilde{z}\tilde{z}}(\tau) = e^{i2\Theta}C_{zz}(\tau)$. This shows that information regarding the directionality of variability must reside in $C_{zz}(t,\tau)$ and not in $R_{zz}(t,\tau)$. If the process is *isotropic*, meaning that its statistics are independent of the rotation angle $\Theta$, then clearly $C_{zz}(t,\tau)$ must vanish."

7. *Page 9, line 20: please give the definition of the Matérn process and formula for its spectrum here. Is it a Gaussian process?*
   This discussion on memory vs. diffusiveness has been split. The high-level discussion is at the end of Section 2.3, while the detailed discussion and table have been moved to Appendix C. Thus, no quantities need to be discussed before they are introduced in the main text. In the appendix, all relevant quantities are referred to by equation number, so that the reader can find their definitions.

8. *Page 9, line 21: in what sense the derivative is taken?*
   This section, which has been moved to the new Appendix C, has been rewritten in such a way that we avoid needing to take derivatives. Instead, the various processes are defined in terms of their transfer functions, the Fourier transforms of their Green's functions in a stochastic integral representation.

9. *Page 10, line 17: what is periodic domain?*
   This is a standard term in geophysical numerical modeling. Its meaning has now been made clear as follows: "A doubly periodic domain means that the $x$-axis is periodic, such that structures passing eastward across the eastern boundary return on the western boundary, and that the $y$-axis is similarly periodic."

10. *Page 13, line 11: please write the range of the parameters.*
    This is now done. "The second is a power-law spectrum that arises for fractional Brownian motion for $\alpha$, termed the *slope parameter*, in the range $1/2 < \alpha < 3/2$.

For the slope parameter $\alpha > 1/2$, the third spectrum is that of a type of random process known as a Matérn process, which we will show to be a *damped* version of fractional Brownian motion, with $\lambda > 0$ playing the role of an inverse damping timescale."

11. *Page 15, line 23: the equality sign here without the reference to Section 3.3 is misleading. In Section 2.3 you define the spectrum for stationary process. Fractional Brownian motion is not stationary, so at that stage it is not clear what this equality means.*
    Good point. We would like to introduce the power law spectrum, and later discuss its meaning in more detail. Therefore, at this location we have added the statement: "While the spectrum of fBm is not defined in the usual sense due to its nonstationary, an expanded version of the notion of a spectrum, discussed in Section 3.3, is found to yield for fBm the form (Flandrin, 1989; Solo, 1992) ... " Furthermore, we have introduced a new symbol, $\widetilde{S}(\omega)$, to distinguish this 'spectrum' from the usual Fourier spectrum $S(\omega)$.

12. *Pages 18-19, beginning of Section 3.3. it is worth mentioning that the spectrum of stationary process and its time-independent spectrum coincide. Thus, time-independent spectrum is a more general notion than spectrum. Otherwise it is not clear, why one can compare the spectrum of stationary process and time-independent spectrum of non-stationary process.*
    A very good point, we have done so. In fact, this suggested to us a reorganization of Section 3.3. to make this point clearer. Please see the first three paragraphs in that section. Again, thank you.

13. *Page 24, lines 13-14: the sentence is not clear. There are two separate integrals in equation (44). Why the integral in (47) cannot be split into two integrals?*
    This has been clarified in the text as follows: "Note that the two components of (47) cannot be written as separate integrals, because writing them as two separate integrals would mean that two different realizations of $dW(s)$ are involved. The two terms in (47) must be based on the *same* realization of $dW(s)$ in order to achieve the initial condition $z(0) = 0$; this is not true for the two terms in (44), which correspond to two different intervals of integration."

14. *Page 25, lines 26-27: the following statement is not clear. 'The first is that there is no upper bound on $\alpha$, so processes can become still smoother than the $\alpha = 3/2$ case that defines the upper limit of the slope parameter for fBm'. If $\alpha M > \alpha fBm$, where $\alpha M$ is the smoothness parameter of the Matérn process and $\alpha fBm$ is the smoothness parameter of the fBm, it does not mean that the corresponding Matérn process is smoother than the corresponding fBm.*

   An interesting and subtle point. It depends on what one means by 'smooth'. This sentence has been changed to read "The first is that there is no upper bound on $\alpha$, so the spectral decay can become even steeper than for the $\alpha = 3/2$ case that defines the upper limit of the slope parameter for fBm."

15. *It would be interesting to have the comparison of your simulation algorithm with known approximate algorithms (for example, spectral method). Note, that exact realizations for some values of $\alpha$ can be generated fast using a modification of circulant embedding called cut-off circulant embedding, see ().*

   We agree, such a comparison would be very interesting. In fact it is already being undertaken by another group (S. Keating, Univ. of New South Wales, pers. comm.). Thank you for the mention of 'cut-off circulant embedding', we were not aware of this. It would also be interesting to know if our Green's function method can be readily generalized to larger classes of processes. As generation methods are not the main focus here, the comparison here is limited to comparing the $O(N^3)$ Cholesky method with the $O(N \log N)$ Green's function method in Figure 8. At the end of Section 5, we have therefore added the sentence "A more detailed comparison between the Green's function method of generation, and other methods such as circulant embedding (Dietrich and Newsam, 1997;

Percival, 2006), is outside the scope of this paper, and is a natural direction for further work."

16. *Please explain how to choose parameters $k$ and $\hat{N}$ optimally. How can one estimate the error of the simulation for given $k$ and $\hat{N}$ ?*

    As we state in Section 5.4 in the text, "If desired, the matrix $R_{m,n}$ in (98) can be computed in order to explicitly check the errors in computing the covariance matrix." To this we have now added: "Thus error can be computed by comparing the difference between the true discretely sampled autocovariance matrix $R_{m,n}$ and the autocovariance matrix $\widehat{R}_{m,n}$ that is satisfied by the process generated through the Green's function method. While this is numerically expensive, it need only be computed one time for a given set of parameters $\alpha$, $\lambda$, $N$, $k$, and $\epsilon$." In addition, we now discuss the default settings in our numerical implementation, which involve making sensible choices for $\hat{N}$ and $k$: "In the numerical implementation described in Appendix A, we set $\epsilon = 0.01$, such that $T_\epsilon$ gives the time at which the time-integrated Green's function reaches one percent of its total time-integrated magnitude. We also set the oversampling parameter $k$ such that there will be at least 10 points per damping timescale $\lambda^{-1}$, which is accomplished by choosing $k = \text{ceil}\,(10 \times \lambda\Delta)$ since $1/(\lambda\Delta)$ is the number of sampled points in one damping timescale. These settings are observed to give fast but accurate performance for a broad range of parameters."

17. *Please explain why you do not use classical maximum likelihood estimation in the time domain. What is the benefit of using frequency-domain maximum likelihood? It would be interesting to see the box-plots and bootstrap confidence intervals for the estimated parameters of the times series generated by the proposed algorithm.*

    Please now find the following paragraph in Section 6.2: "A standard approach would be to form a parametric estimate using the maximum likelihood method implemented in the time domain. However, this method involves a computationally

expensive matrix inversion, which becomes a limiting factor when analyzing large datasets. An alternative approach to estimating the parameters is to do so in the frequency domain using a method called the Whittle likelihood (Whittle, 1953). This approach is considerably faster than time-domain maximum likelihood, with $O(N \log N)$ versus $O(N^2)$ behavior, yet is known to give approximately the same results. It also has the advantage of letting us only fit the parametric model over a specified band of frequencies. The Whittle likelihood method proceeds as follows."

18. *For the reader's convenience, when citing books, please include the number of the relevant chapter or pages.*
Page or chapter numbers have now been given for the half-dozen or so books for which we had previously neglected them.

Minor issues and typos

1. *Page 3, line 2: 'a uniform rotation rate to is shown'*
Fixed.

2. *Page 3, line 30: 'This paper was inspired by the need to a develop'*
Fixed, thank you.

3. *Page 9, Table 1: "The term in the box is the Matérn process", you probably mean that the term in the box is the spectrum of the Matérn process. Please write the ranges for $\lambda$, $\Omega$, $\alpha$.*
Both of these have been changed as requested.

4. *Page 9, line 17: if a function is absolutely integrable, then it is integrable; not vice versa.*

Thank you very much for noticing this error. Fortunately, it was only a typographic error, and the reasoning in the rest of the paragraph holds.

5. *Page 17, lines 5-6: the sentence "The exponent $2\alpha - 1$ varies from ..." is very difficult to understand, however it has a very simple meaning, namely $0 < 2\alpha - 1 < 2$.*
   This has been replaced with 'The exponents take on values in the range $0 < 2\alpha - 1 < 2$ due to the fact that $1/2 < \alpha < 3/2$,' which we agree is clearer, thank you.

6. *Page 19, line 3: 'faveraging'*
   Fixed, thank you.

7. *Page 27, lines 1-3: the sentence is unclear. What vanishes at $|\omega| = \lambda$?*
   This was quite unclear and has been clarified. The point is simply that the fractional rate of decrease of the spectrum achieves a maximum at this frequency.

8. *Page 41, line 2: we proceed as values*
   This has been corrected to read 'we proceed as follows.'

9. *Page 41, line 16: turbulence turbulence.*
   Fixed.